# Optimizing Agentic Reasoning with Retrieval via Synthetic Semantic Information Gain Reward

**Senkang Hu** [1 2]   **Yong Dai** [3]   **Yuzhi Zhao** [4]   **Yihang Tao** [1 2]   **Yu Guo** [1 2]   **Zhengru Fang** [1 2]   **Sam Kwong** [5]   **Yuguang Fang** [1 2]

## Abstract

Agentic reasoning enables large reasoning models (LRMs) to dynamically acquire external knowledge, yet optimizing the retrieval process remains challenging due to the lack of dense, principled reward signals. In this paper, we introduce *InfoReasoner*, a unified framework that incentivizes effective information seeking via a *synthetic semantic information gain reward*. Theoretically, we redefine information gain as uncertainty reduction over the model's belief states, establishing key properties including non-negativity, telescoping additivity, and channel monotonicity. Practically, to enable scalable optimization without manual intermediate retrieval annotations, we instantiate this principle as a semantic information gain reward computed from the model's output distributions using *semantic clustering via bidirectional textual entailment*. This training reward provides dense credit for retrieval steps while remaining anchored to final-answer correctness, enabling efficient training via Group Relative Policy Optimization (GRPO). Experiments on seven question-answering benchmarks, MATH500, and WebDetective show consistent gains over strong retrieval-augmented baselines, supporting our dense semantic information gain as a practical training signal for agentic retrieval.

## 1. Introduction

Large reasoning models (LRMs) have demonstrated remarkable capabilities in complex problem-solving by integrating extended reasoning chains with external knowledge retrieval (Guo et al., 2025; Jin et al., 2025a; Li et al., 2025a). This *agentic reasoning* paradigm enables models to dynamically retrieve relevant information during inference, combining the parametric knowledge of language models with the factual accuracy of external knowledge bases. However, optimizing such retrieval-augmented reasoning systems remains a fundamental challenge: *how should we measure and reward the value of each retrieval action* to guide the agent toward more effective information gathering and reasoning?

Existing approaches to optimizing agentic reasoning with retrieval face several critical limitations. First, most methods rely on *supervised fine-tuning* with human-annotated demonstrations (Li et al., 2025a; Jin et al., 2025a), which limits scalability and fails to capture the nuanced value of retrieval actions in open-ended reasoning scenarios. Second, reinforcement learning (RL) methods that optimize retrieval directly often employ *task-specific rewards* (e.g., final answer correctness) (Xiong et al., 2025; Tan et al., 2025), which suffer from signal sparsity, feedback delay, and inability to distinguish between retrieval actions that are equally distant from the final answer. Third, while some works attempt to incorporate intermediate reasoning signal quality (Zhang et al., 2025c), they rely on heuristic process supervision and lack a formal probabilistic framework that mathematically guarantees local retrieval decisions contribute to the global reduction of epistemic uncertainty.

To address these limitations, we propose a principled approach that quantifies the intrinsic value of each retrieval step. The core insight underlying our approach is that *effective retrieval should reduce an agent's uncertainty over the correct answer*. Intuitively, a good retrieval action is one that concentrates an agent's belief distribution over possible answers, while a poor action either fails to reduce uncertainty or, worse, increases confusion. This perspective naturally connects retrieval optimization to the well-established concept of *information gain* from information theory and active learning. However, traditional information gain metrics require access to dense ground-truth annotations or oracle belief states, which are unavailable in practical agentic reasoning scenarios where an agent must learn from its own generated outputs.

---

[1]Hong Kong JC STEM Lab of Smart City. [2]City University of Hong Kong. [3]Fudan University. [4]Huazhong University of Science and Technology. [5]Lingnan University. Correspondence to: Yuguang Fang <my.Fang@cityu.edu.hk>.

*Proceedings of the $43^{rd}$ International Conference on Machine Learning*, Seoul, South Korea. PMLR 306, 2026. Copyright 2026 by the author(s).

In this paper, we propose *InfoReasoner*, a unified framework that addresses these challenges by redefining information gain as *uncertainty reduction over belief states* and designing a *synthetic semantic information gain reward* that can be computed from the model's own output distributions without requiring manual intermediate retrieval annotations. Our theoretical contribution establishes information gain as a principled reward signal by proving key properties: *non-negativity* (information gathering never hurts), *telescoping additivity* (local gains accumulate to global uncertainty reduction), and *monotonicity* (better information channels yield higher gains). These properties ensure that optimizing per-step information gain aligns with the long-term goal of reducing epistemic uncertainty about the final answer.

To make this framework practical, we introduce an output-aware reward estimator that computes information gain directly from semantic equivalence classes of the model's generated outputs. Specifically, we: 1) sample multiple answer sequences under different conditioning contexts (with and without retrieved evidence), 2) cluster semantically equivalent answers using bi-directional textual entailment, 3) estimate belief distributions over semantic classes, and 4) compute a gold-class gain reward that measures whether retrieved evidence increases probability mass on the correct semantic class. This reward gives dense supervision to intermediate retrieval steps without requiring manual retrieval annotations, while the final-answer EM reward anchors training to task correctness.

We optimize retrieval policies using this reward signal along with the output reward through Group Relative Policy Optimization (GRPO) (Shao et al., 2024), which efficiently estimates advantages without requiring explicit value functions. Our experiments on seven question-answering benchmarks, tool-integrated and long-horizon agentic settings demonstrate that InfoReasoner consistently improves reasoning accuracy over strong retrieval-augmented baselines, achieving up to 5.4% average improvement while remaining computationally stable.

**Contributions.** Our main contributions are: 1) a theoretical framework that redefines information gain as uncertainty reduction over belief states, establishing formal guarantees for its use as a reward signal; 2) a practical, output-aware training reward for computing synthetic semantic information gain directly from model outputs without manual retrieval annotations; 3) empirical validation demonstrating consistent improvements across diverse reasoning benchmarks; and 4) a scalable optimization approach that enables efficient policy learning for agentic reasoning with retrieval.

## 2. Rethinking Information Gain as Uncertainty Reduction over Belief States

Optimizing intermediate retrieval steps is challenging due to the sparse and delayed nature of final answer rewards. To address this, before detailing our practical implementation in Section 3, we first establish the theoretical foundations of InfoReasoner. We model the reasoning process as a Partially Observable Markov Decision Process (POMDP) to formally define "uncertainty" in the context of agentic search. Crucially, this theoretical framework motivates our design choices in Section 3. Specifically, the abstract *"belief state"* derived here directly maps to the *"semantic cluster distribution"* we construct later, and the *"uncertainty reduction"* properties proved here (Non-negativity, Telescoping Additivity) justify using our proposed intrinsic reward as a dense training signal.

First, we formulate agentic reasoning with retrieval as a POMDP:

$$\mathcal{M} = (\mathcal{S}, \mathcal{A}, \mathcal{O}, T, \Omega, \gamma), \tag{1}$$

where $\mathcal{S}$ is the (latent) state space, which may not be observed directly. $\mathcal{A}$ is the action space (e.g., generating a reasoning step or issuing a retrieval query). $\mathcal{O}$ is the observation space (e.g., retrieved documents or tool outputs). $T(s' \mid s, a)$ is the state transition probability, which means the probability of transitioning from state $s$ to state $s'$ after taking action $a$. $\Omega(o \mid s', a)$ is the observation probability, which means the probability of observing $o$ after taking action $a$ and transitioning to state $s'$. $\gamma \in [0, 1]$ is the discount factor.

### 2.1. Bayesian Belief Update

**Definition 2.1** (Belief State). Since the environment state $s \in \mathcal{S}$ may not be directly observable, the agent cannot know the true state with certainty. Therefore, we define a *belief state* $b_t$ as the posterior probability distribution over a latent task variable $Y$:

$$b_t(y) = P(Y = y \mid o_{\leq t}), \tag{2}$$

where $Y$ is the latent task variable (e.g., the correct answer or its semantic equivalence class), $o_{\leq t}$ is the observation history (including retrieved documents or tool outputs), and $b_t$ is the belief state, which is a probability distribution reflecting the agent's uncertainty about the correct answer $Y$ after seeing all historical observations.

**Remark:** Computing the exact belief state $b_t$ over the infinite space of natural language sequences is intractable. In Section 3, we approximate the belief state $b_t$ as a probability distribution over *semantic equivalence classes* of generated answers, estimated via sampling and clustering.

**Definition 2.2** (Bayes Belief Update). In the context of LLM-based reasoning agents, the belief update must ac-

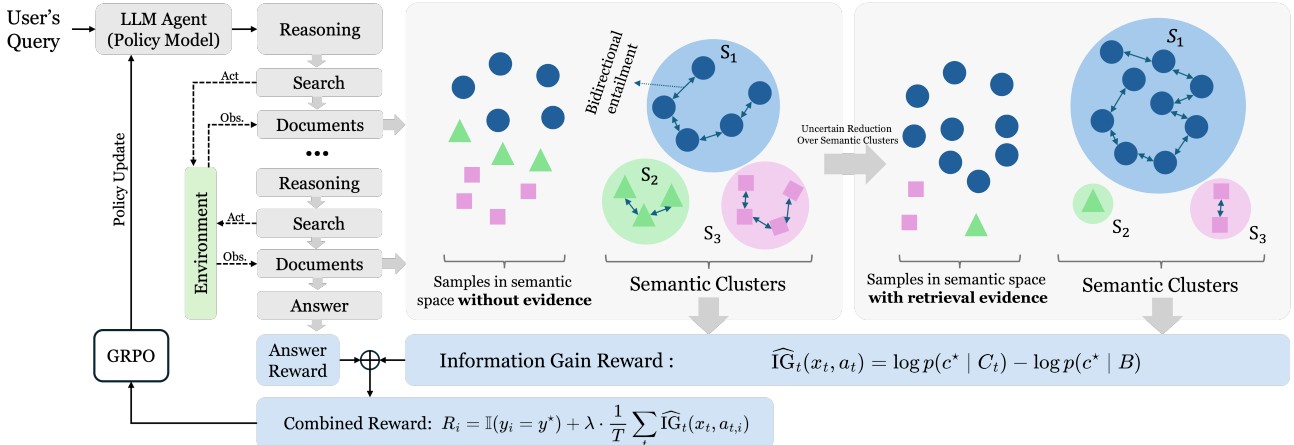

*Figure 1.* Overview of **InfoReasoner**. The framework estimates the agent's *belief state* by sampling candidate answers and grouping semantically equivalent ones. It then calculates an *Information Gain* intrinsic reward by measuring the reduction in semantic uncertainty (entropy) when retrieved evidence is provided compared to a retrieval-free baseline, thereby incentivizing the agent to acquire uncertainty-resolving information.

count for the fact that observations are generated through a two-stage process involving both environmental retrieval and LLM generation. The agent takes action $a_t$ (e.g., retrieval), and the environment returns evidence $e_t \sim P(\cdot \mid Y = y, a_t)$. The LLM then generates an observation $O_t \sim P_{\text{LLM}}(\cdot \mid e_t, b_t)$. Assuming the LLM's interpretation of evidence dominates prior bias (i.e., $P_{\text{LLM}}(O_t \mid e_t, b_t) \approx P_{\text{LLM}}(O_t \mid e_t)$, see Appendix B.1 for details), the belief update reduces to:

$$b_{t+1}(y) = \frac{P(O_t \mid Y = y, a_t)\, b_t(y)}{\sum_{y' \in \mathcal{Y}} P(O_t \mid Y = y', a_t)\, b_t(y')}. \quad (3)$$

This simplified version is used in our subsequent analysis for tractability.

## 2.2. Uncertainty Functional

To quantify an agent's epistemic uncertainty about the latent answer $Y$, we introduce an uncertainty functional $\mathcal{U}$.

**Definition 2.3** (Uncertainty Functional). We define an *uncertainty functional* $\mathcal{U}$ as a mapping

$$\mathcal{U} : \mathcal{B} \to \mathbb{R}_{\geq 0}, \qquad b \mapsto \mathcal{U}(b), \quad (4)$$

where $\mathcal{B}$ denotes the space of all possible belief states, that is, the set of all probability distributions $b$ over the latent variable $Y$ with support $\mathcal{Y}$, so that each $b(y) = P(Y = y \mid \text{history})$. The mapping $\mathcal{U}$ is a functional that assigns to each belief state $b \in \mathcal{B}$ a non-negative real value $\mathcal{U}(b)$, which quantifies the uncertainty associated with that belief. The notation $b \mapsto \mathcal{U}(b)$ simply emphasizes that the input is a probability distribution $b$ and the output is its corresponding uncertainty value. The functional $\mathcal{U}$ satisfies the following axioms:

1. *Minimality:* If $b$ is degenerate (assigns probability 1 to some $y$), then $\mathcal{U}(b) = 0$.

2. *Concavity:* For any $b_1, b_2 \in \mathcal{B}$ and $\lambda \in [0, 1]$,

$$\mathcal{U}(\lambda b_1 + (1 - \lambda) b_2) \geq \lambda \mathcal{U}(b_1) + (1 - \lambda) \mathcal{U}(b_2). \quad (5)$$

3. *Expected Monotonicity:* For any belief $b$ and action $a$,

$$\mathbb{E}_{O \sim P(\cdot \mid b, a)} \big[ \mathcal{U}(b_{t+1}) \big] \leq \mathcal{U}(b_t). \quad (6)$$

These axioms state that $\mathcal{U}$ quantifies epistemic uncertainty: it is zero under certainty, concave under mixture, and cannot increase in expectation after incorporating new information. A typical example is the Shannon entropy: $\mathcal{U}_H(b) \triangleq -\sum_{y \in \mathcal{Y}} b(y) \log b(y)$.

**Remark:** In Section 3, we instantiate $\mathcal{U}$ using the *semantic entropy* of the belief distribution over semantic classes.

## 2.3. Information Gain as Uncertainty Reduction

Having defined the uncertainty functional $\mathcal{U}$ to quantify an agent's epistemic uncertainty about the latent answer $Y$, we can now formalize the concept of information gain. Intuitively, information gain represents the reduction in this uncertainty resulting from the acquisition of new information. This leads to the following definitions.

**Definition 2.4** (One-step Information Gain). Given $\mathcal{U}$, the *realized information gain* at time $t$ is

$$\text{IG}_t = \mathcal{U}(b_t) - \mathcal{U}(b_{t+1}), \quad (7)$$

where $b_{t+1}$ is the posterior belief obtained from $b_t$ after taking action $a_t$ and observing $O_t$ via the Bayes update. In other words, $\text{IG}_t$ quantifies how much the agent's uncertainty has decreased, or its certainty has increased, after taking such action when observing the system is at state $O_t$.

**Remark:** This theoretical quantity motivates the reward design in Section 3. In practice, we distinguish the entropy-

based semantic information gain from the gold-class gain reward actually optimized during training.

**Definition 2.5** (Expected Information Gain). The *expected information gain* of action $a$ under belief $b$ is

$$\text{EIG}(a \mid b) = \mathbb{E}_{O \sim P(\cdot \mid b,a)} \Big[ \mathcal{U}(b) - \mathcal{U}(b_{t+1}) \Big]. \quad (8)$$

This value represents the average reduction in uncertainty that an agent *expects* to achieve by taking action $a$. It serves as a crucial forward-looking metric for decision-making, allowing the agent to choose actions that are most likely to resolve its uncertainty about the final answer.

**Proposition 2.6** (Non-negativity under Ideal Updates). *If $U$ satisfies the Expected Non-Increase axiom described in Eq. (6), then for any belief $b$ and action $a$, under the assumption of consistent Bayesian belief updates, the Expected Information Gain is non-negative:*

$$\text{EIG}(a \mid b) \geq 0, \quad (9)$$

*with equality iff $O \perp Y \mid (b,a)$. This theoretical lower bound serves as the optimality condition for our agentic reasoning policy.*

**Remark (Theoretical vs. Realized Gain):** It is important to distinguish between the *theoretical* expected gain (which is non-negative for a rational agent) and the *realized* gain $\text{IG}_t$ observed during training. In practice, LLM agents are not perfect Bayesian updaters. A retrieval action returning misleading or "poisoned" context can increase the entropy of the model's belief state, resulting in a *negative realized information gain* ($\text{IG}_t < 0$). Crucially, within our RL framework, this is not a failure mode but a *desirable penalty signal*. A negative reward discourages the policy from executing retrieval actions that confuse the model or retrieving documents that conflict with the reasoning chain, effectively aligning the agent's behavior with the theoretical optimality derived in Proposition 2.6. The proof is given in Appendix B.2.

**Proposition 2.7** (Telescoping Additivity). *Along any belief trajectory $b_0 \to b_1 \to \cdots \to b_T$ generated by the agent,*

$$\sum_{t=0}^{T-1} \text{IG}_t = \mathcal{U}(b_0) - \mathcal{U}(b_T). \quad (10)$$

*Thus, local information gains telescope to the global reduction in uncertainty.*

Telescoping additivity is a crucial property that bridges local, step-wise rewards with the global task objective. It demonstrates that, under the ideal belief-update formulation, local entropy-based gains accumulate to global uncertainty reduction. This property motivates step-level retrieval credit assignment, while the practical reward in Section 3

uses a supervised gold-class surrogate to improve alignment with final-answer correctness. The proof is given in Appendix B.3.

**Proposition 2.8** (Monotonicity w.r.t. Information Channels). *If action $a_1$ induces an observation channel that Blackwell-dominates $a_2$, then*

$$\text{EIG}(a_1 \mid b) \geq \text{EIG}(a_2 \mid b), \quad \forall b. \quad (11)$$

This proposition ensures that our framework behaves rationally when comparing different information-gathering actions. It guarantees that an action leading to a more informative outcome (e.g., querying a more reliable knowledge base, using a more precise tool) will be assigned an equal or higher EIG value. This is essential for decision-making, as it directs an agent to systematically prefer higher-quality information sources, thereby optimizing its retrieval and reasoning strategy. The proof is given in Appendix B.4.

### 2.4. Interpretation

This framework establishes that *information gain equals uncertainty reduction* of a generalized functional $\mathcal{U}$. The telescoping property (Proposition 2.7) explains why local uncertainty reduction is a meaningful retrieval objective under ideal belief updates, while monotonicity (Proposition 2.8) gives a rational preference ordering over information sources. Section 3 then turns this principle into a tractable semantic information gain reward that preserves the before and after retrieval contrast while anchoring the signal to correctness.

## 3. Method

To operationalize the theoretical framework in Section 2 into a tractable algorithm, we instantiate the abstract belief state $b_t$ as a probability distribution over *semantic equivalence classes* of sampled answers, and instantiate the uncertainty functional $\mathcal{U}$ as the *semantic entropy* of this distribution. We then compute $\text{IG}_t$ as the reduction in semantic entropy after incorporating retrieved evidence, and use it as an intrinsic reward signal. The remainder of this section details our semantic clustering procedure for belief estimation and the resulting information-gain reward computation. The overall framework is illustrated in Fig. 1.

### 3.1. Semantic Clustering via Bidirectional Textual Entailment

A key innovation of our framework is the estimation of semantic uncertainty through clustering model outputs into equivalence classes. Traditional approaches using token overlap or embedding similarity fail to capture semantic equivalence across diverse answer formulations. Consider the answers "Einstein" and "He is Albert Einstein"; these are semantically identical but syntactically distinct.

To overcome these challenges, inspired by textual entailment (Androutsopoulos & Malakasiotis, 2010), we introduce a robust semantic clustering algorithm based on bidirectional textual entailment. At step $t$, the policy LLM $\pi_\theta$ receives problem statement $x_t$ and belief context from $b_t$. It emits a retrieval action $a_t$ (query/search), the retriever returns retrieved evidence $e_t$, and the model produces observation $O_t$. We compare two conditioning contexts for uncertainty: 1) $Z = B(x_t)$: a fact-free paraphrase of $x_t$ that preserves only task framing (e.g., query and system prompt). 2) $Z = C_t(a_t) = x_t \oplus e_t$, where $x_t$ is concatenated with retrieved evidence $e_t$ via the retrieval action $a_t$. Both are fed to the same $\pi_\theta$. Then, we sample $M$ sequences $\{s_1^{(Z)}, \ldots, s_M^{(Z)}\} \sim \pi_\theta(\cdot \mid x_t, Z)$ from the policy for context $Z \in \{B(x_t), C_t(a_t)\}$. We define semantic equivalence as follows.

**Definition 3.1** (Semantic Equivalence via Bidirectional Entailment)**.** Two answer sequences $s_i$ and $s_j$ belong to the same semantic class if and only if they mutually entail each other in the context of question $x_t$:

$$s_i \leftrightarrow s_j \iff$$
$$P_{\text{NLI}}(s_i \vDash s_j \mid x_t) > \tau \text{ and } P_{\text{NLI}}(s_j \vDash s_i \mid x_t) > \tau \qquad (12)$$

where $\vDash$ denotes the entailment relation (i.e., $s_i \vDash s_j$ means $s_i$ logically entails $s_j$), $P_{\text{NLI}}$ is a pretrained natural language inference (NLI) model that outputs the probability of entailment between two text sequences given a context, and $\tau$ is a confidence threshold.

This definition captures the intuitive notion that two answers are equivalent if each logically implies the other given the question context. We partition sequences into semantic classes $\mathcal{C}$ by constructing an undirected graph (sequences as nodes, edges for bidirectional entailment) and extracting connected components. This discrete distribution over $\mathcal{C}$ approximates the belief state $b_t$ in Eq. (2). The details are given in Algorithm 1.

### 3.2. Synthesizing Information Gain Reward via Semantic Uncertainty Reduction

With semantic classes established, the belief distribution over these classes for each context can be estimated by the accumulated sequence likelihoods within each class. For context $Z \in \{B(x_t), C_t(a_t)\}$, the belief distribution $p(c \mid Z)$ over semantic classes $c$ can be formulated as:

$$p(c \mid Z) = \sum_{s \in c} p_\theta(s \mid Z),$$
$$= \sum_{s \in c} \prod_j p_\theta(s_j \mid s_{<j}, Z). \qquad (13)$$

where $s_j$ is the $j$-th output token and $s_{<j}$ is the token sequence. The semantic belief under $Z$ can be estimated

by the entropy of the belief distribution over all semantic classes:

$$H_{\text{sem}}(Z) = -\sum_{c \in \mathcal{C}} p(c \mid Z) \log p(c \mid Z), \qquad (14)$$

Following the definition of information gain in Eq. (7), we instantiate $\mathcal{U}$ with semantic entropy in Eq. (14), which yields the following full-distribution, entropy-based semantic information gain estimator:

$$\widehat{\text{IG}}_t^{\text{ent}}(x_t, a_t) = H_{\text{sem}}(x_t, B) - H_{\text{sem}}(x_t, C_t), \qquad (15)$$

By construction, $\widehat{\text{IG}}_t^{\text{ent}}$ measures aggregate uncertainty reduction over all semantic classes. This matches the theoretical uncertainty-reduction viewpoint, yet it can still assign positive reward when the model becomes confidently wrong if its posterior collapses onto an incorrect semantic class.

To provide a more direct learning signal that focuses on the correct answer, we consider the model's belief distribution over the golden answer $y^\star$. We identify the correct semantic class $c^\star$ as the unique class containing sequences semantically equivalent to $y^\star$:

$$c^\star = \arg\max_{c \in \mathcal{C}} \mathbb{I}(c, y^\star), \qquad (16)$$

where $\mathbb{I}(c, y^\star) = 1$ if the semantic meaning of $c$ is equivalent to $y^\star$ (semantically equivalent via bidirectional entailment), otherwise 0. The information gain reward term is then computed using the probability of the correct class $p(c^\star \mid x_t, C_t)$ (as defined in Eq. (13)) under different contexts:

$$r_t(x_t, a_t) = \log p(c^\star \mid C_t) - \log p(c^\star \mid B). \qquad (17)$$

This formulation rewards the policy for increasing the probability mass on the correct semantic class after retrieving evidence, providing a direct signal that guides retrieval toward information that supports the correct answer. Equivalently, Eq. (17) is a gold-class uncertainty reduction objective with $U_{\text{gold}}(Z) = -\log p(c^\star \mid Z)$, since $r_t = U_{\text{gold}}(B) - U_{\text{gold}}(C_t)$. In the rest of the paper, $\widehat{\text{IG}}_t^{\text{ent}}$ denotes the theoretical entropy-based quantity, while $r_t$ denotes the practical reward optimized during training.

### 3.3. Policy Optimization with Information Gain Reward

To optimize the retrieval policy $\pi_\theta$, we employ Group Relative Policy Optimization (GRPO) (Shao et al., 2024), which estimates advantages from a group of sampled outputs for the same input, reducing training instability in sparse reward settings.

The total reward $R_i$ for each output integrates both the final outcome correctness and the intermediate information gain.

*Table 1.* Accuracy comparison of our method versus baseline methods with Qwen2.5-3B across various QA benchmarks. Bold denotes best results, and underline denotes second best results.

| Methods | Single-Hop QA | | | Multi-Hop QA | | | | Avg. |
|---|---|---|---|---|---|---|---|---|
| | NQ | TriviaQA | PopQA | HotpotQA | 2Wiki | MuSiQue | Bamboogle | |
| *w/o Retrieval* | | | | | | | | |
| Direct Generation | 0.106 | 0.288 | 0.108 | 0.149 | 0.244 | 0.020 | 0.024 | 0.134 |
| SFT | 0.249 | 0.292 | 0.104 | 0.186 | 0.248 | 0.044 | 0.112 | 0.176 |
| R1-Instruct (Guo et al., 2025) | 0.210 | 0.449 | 0.171 | 0.208 | 0.275 | 0.060 | 0.192 | 0.224 |
| R1-Base (Guo et al., 2025) | 0.226 | 0.455 | 0.173 | 0.201 | 0.268 | 0.055 | 0.224 | 0.229 |
| CoT | 0.048 | 0.185 | 0.054 | 0.092 | 0.111 | 0.022 | 0.232 | 0.106 |
| *w/ Retrieval (3B)* | | | | | | | | |
| RAG (Lewis et al., 2020) | 0.348 | 0.544 | 0.387 | 0.255 | 0.226 | 0.047 | 0.080 | 0.270 |
| Search-o1 (Li et al., 2025a) | 0.238 | 0.472 | 0.262 | 0.221 | 0.218 | 0.054 | **0.320** | 0.255 |
| IRCoT (Trivedi et al., 2023) | 0.111 | 0.312 | 0.200 | 0.164 | 0.171 | 0.067 | 0.240 | 0.181 |
| Search-R1-3B-Base (Jin et al., 2025a) | 0.394 | 0.580 | 0.390 | 0.292 | 0.260 | 0.048 | 0.108 | 0.296 |
| Search-R1-3B-Instruct (Jin et al., 2025a) | 0.405 | 0.566 | 0.354 | 0.316 | 0.224 | 0.056 | 0.184 | 0.301 |
| InForge-3B (Qian & Liu, 2025) | 0.386 | 0.504 | 0.374 | 0.236 | 0.114 | 0.060 | 0.272 | 0.278 |
| AutoRefine-3B-Base (Shi et al., 2025) | 0.281 | 0.428 | 0.272 | 0.208 | 0.162 | 0.036 | 0.136 | 0.217 |
| AutoRefine-3B-Instruct (Shi et al., 2025) | 0.383 | 0.522 | 0.344 | 0.222 | 0.118 | 0.022 | 0.004 | 0.231 |
| *Ours* | | | | | | | | |
| InfoReasoner-3B | **0.453** | **0.634** | **0.442** | **0.344** | **0.324** | **0.080** | 0.144 | **0.346** |
| *w/ Retrieval (7B)* | | | | | | | | |
| Search-R1-7B-Base (Jin et al., 2025a) | 0.391 | 0.556 | 0.364 | 0.324 | 0.202 | 0.076 | 0.328 | 0.320 |
| Search-R1-7B-Instruct (Jin et al., 2025a) | 0.407 | 0.590 | 0.390 | 0.340 | 0.194 | 0.080 | 0.360 | 0.337 |
| ReSearch-7B-Base (Chen et al., 2025) | 0.268 | 0.446 | 0.270 | 0.232 | 0.172 | 0.062 | 0.240 | 0.241 |
| ReSearch-7B-Instruct (Chen et al., 2025) | 0.333 | 0.568 | 0.354 | 0.334 | 0.190 | 0.080 | 0.312 | 0.310 |
| ReasonRAG-7B (Zhang et al., 2025c) | 0.294 | 0.578 | 0.348 | 0.330 | 0.244 | 0.080 | 0.344 | 0.318 |
| AutoRefine-7B-Base (Shi et al., 2025) | 0.439 | 0.608 | 0.402 | 0.410 | 0.242 | 0.116 | 0.368 | 0.369 |
| Ours | | | | | | | | |
| InfoReasoner-7B | **0.447** | **0.614** | **0.416** | **0.414** | **0.302** | **0.120** | **0.424** | **0.391** |

Specifically, it is a weighted sum of the task-specific exact match score and the cumulative information gain reward:

$$R_i = \mathbb{I}(y_i = y^\star) + \lambda \cdot \frac{1}{T_i} \sum_t r_t(x_t, a_{t,i}), \qquad (18)$$

where $\mathbb{I}(y_i = y^\star)$ is the exact match indicator function between the predicted answer $y_i$ and the ground truth $y^\star$, and $r_t$ is the estimated information gain gain reward for retrieval action $a_{t,i}$ at step $t$. The hyperparameter $\lambda$ balances the incentive for accurate reasoning with the intermediate supervision for informative retrieval. This composite reward structure encourages the policy to learn strategies that are both accurate and information-efficient.

## 4. Experiments

### 4.1. Experimental Setup

**Datasets.** Our evaluation covers seven question answering (QA) datasets. For single-hop QA, we use Natural Questions (NQ) (Kwiatkowski et al., 2019), TriviaQA (Joshi et al., 2017), and PopQA (Mallen et al., 2023). For multi-hop QA, the benchmarks are HotpotQA (Yang et al., 2018), 2Wiki-MultiHopQA (2Wiki) (Ho et al., 2020), MuSiQue (Trivedi et al., 2022), and Bamboogle (Press et al., 2023). These datasets represent a broad spectrum of search and reasoning tasks, facilitating thorough evaluation. Additionally, our model is trained using the training splits of NQ and Hot-potQA, and evaluated on the test sets of these seven QA datasets. To test generalization beyond short-answer QA, we evaluate on MATH500 (Hendrycks et al., 2021) for tool-integrated mathematical reasoning and WebDetective (Song et al., 2025) for long-horizon multi-hop deep search.

**Baseline Methods.** For the seven QA benchmarks, we compare with two categories of baselines: retrieval-augmented baselines and retrieval-free baselines. The retrieval-augmented baselines include vanilla retrieval-augmented generation (RAG), Search-o1, IRCoT, Search-R1-3B-Base (and Instruct), Search-R1-7B-Base (and Instruct), ReSearch-7B-Base (and Instruct), InForge-3B, ReasonRAG-7B, AutoRefine-3B-Base (and Instruct). The retrieval-free baselines include Direct Generation, supervised fine-tuning (SFT), and R1-like training (R1). Implementation details are provided in Appendix D. For tool-integrated reasoning, we compare against ZeroTIR-7B and SimpleTIR-7B. For WebDetective, we compare against GPT-5-Chat, Gemini-2.5-Pro, and DeepSeek-V3.1.

*Table 2.* Ablation study on the information gain coefficient $\lambda$ across seven QA benchmarks. Bold denotes best results, and underline denotes second best results.

| Methods | Single-Hop QA | | | Multi-Hop QA | | | | Avg. |
|---|---|---|---|---|---|---|---|---|
| | NQ | TriviaQA | PopQA | HotpotQA | 2Wiki | MuSiQue | Bamboogle | |
| InfoReasoner ($\lambda = 1.0$) | 0.435 | 0.600 | 0.432 | 0.320 | 0.274 | 0.052 | 0.128 | 0.320 |
| InfoReasoner ($\lambda = 0.8$) | 0.436 | 0.590 | 0.426 | 0.292 | 0.238 | 0.032 | 0.112 | 0.304 |
| InfoReasoner ($\lambda = 0.6$) | **0.452** | **0.634** | **0.442** | **0.344** | **0.324** | **0.080** | **0.144** | **0.346** |
| InfoReasoner ($\lambda = 0.4$) | 0.443 | 0.602 | 0.424 | 0.324 | 0.266 | 0.058 | 0.088 | 0.315 |
| InfoReasoner ($\lambda = 0.2$) | 0.448 | 0.618 | 0.432 | 0.322 | 0.288 | 0.044 | 0.136 | 0.327 |
| InfoReasoner ($\lambda = 0.0$) | 0.426 | 0.538 | 0.426 | 0.294 | 0.254 | 0.040 | 0.118 | 0.299 |

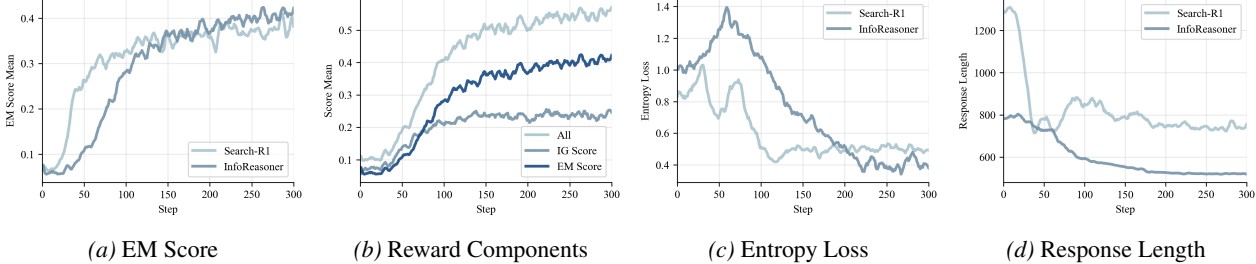

| *(a)* EM Score | *(b)* Reward Components | *(c)* Entropy Loss | *(d)* Response Length |
|---|---|---|---|

*Figure 2.* Training dynamics analysis: (a) EM score trajectories comparison between InfoReasoner and Search-R1; (b) Decomposition of total reward into Information Gain (IG) and EM scores; (c) Entropy loss comparison; (d) Response length comparison between InfoReasoner and Search-R1.

## 4.2. Main Comparative Results

Table 1 presents the comparison between InfoReasoner and various baselines across seven QA benchmarks. InfoReasoner establishes new state-of-the-art performance at both 3B and 7B scales. In the 3B parameter setting, InfoReasoner-3B achieves an average accuracy of 34.6%, significantly outperforming Search-R1-3B-Instruct (30.1%) and the standard RAG model (27.0%). It even surpasses several 7B-scale baselines, such as Search-R1-7B-Instruct (33.7%), demonstrating exceptional parameter efficiency. InfoReasoner-3B achieves the best performance among 3B models on single-hop QA scenarios (NQ, TriviaQA, PopQA) and multi-hop tasks, specifically achieving top results on HotpotQA (34.4%) and 2Wiki (32.4%). Scaling to 7B parameters, InfoReasoner-7B improves the average accuracy to 39.1%, consistently outperforming all 7B baselines, including AutoRefine-7B-Base (36.9%) and Search-R1-7B-Instruct (33.7%), achieving the best results on all evaluated datasets. Beyond the seven short-answer QA benchmarks, we further evaluate InfoReasoner in tool-integrated and long-horizon agentic settings. Table 4 shows that on MATH500 with Python tool execution, InfoReasoner-7B reaches 85.8%, outperforming ZeroTIR-7B (80.2%) and SimpleTIR-7B (84.6%). Table 3 shows that on WebDetective, a long-horizon deep-search benchmark, InfoReasoner-3B improves Pass@1 from 26.5% to 30.0% over Search-R1-3B-Base.

## 4.3. Ablation Study on the Information Gain Coefficient

We conduct an ablation study by varying the information gain coefficient $\lambda$ from 0.0 to 1.0 across all seven QA benchmarks (Table 2). Our experiments reveal that $\lambda = 0.6$ achieves the best overall performance with an average accuracy of 34.6%. When $\lambda = 0.0$ (degenerating to Search-R1), performance drops to 29.9%, demonstrating that the information gain signal is essential and improves performance by 4.7 percentage points. When $\lambda$ is too large ($\lambda = 1.0$ or 0.8), performance degrades to 32.0% and 30.4% respectively, indicating that over-emphasizing information gain harms overall performance. The optimal $\lambda = 0.6$ suggests that *information gain should contribute substantially but not dominate the reward function, balancing epistemic exploration with pragmatic exploitation*.

## 4.4. Training Dynamics Analysis

To further analyze our method, we visualize the evolution of exact match (EM) scores, reward components, entropy loss, and response length during the training process.

**Performance Comparison.** Figure 2a presents the EM score trajectories comparing InfoReasoner ($\lambda = 0.6$) with Search-R1 ($\lambda = 0.0$). InfoReasoner exhibits slower EM score growth in the early training phase compared to Search-R1, because it assigns *positive rewards not only to correct answers but also to retrievals that provide useful information*, even when the final answer is incorrect. This reward structure encourages exploration of diverse retrieval strategies rather than premature convergence to locally optimal

*Table 3.* WebDetective long-horizon search results. Frontier-model results are from the benchmark paper (Song et al., 2025).

| Model | Knowledge Suff. ↑ | Search Score ↑ | Generation Score ↑ | Knowledge Util. ↑ | Forget ↓ | Lead-astray ↓ | Pass@1 ↑ |
|---|---|---|---|---|---|---|---|
| GPT-5-Chat | 0.580 | 0.595 | **0.157** | **0.281** | 0.474 | **0.319** | 0.295 |
| Gemini-2.5-Pro | 0.655 | 0.730 | 0.116 | 0.247 | **0.443** | 0.351 | 0.285 |
| DeepSeek-V3.1 | 0.615 | 0.565 | 0.136 | 0.163 | 0.447 | 0.447 | 0.170 |
| Search-R1-3B-Base | 0.880 | 0.905 | 0.110 | 0.250 | 0.483 | 0.364 | 0.265 |
| InfoReasoner-3B | **0.890** | **0.910** | 0.119 | 0.269 | 0.506 | 0.337 | **0.300** |

---

**Case Study: InfoReasoner**

**Query:** In what city was the band behind the album Love Bites formed?
**Ground-truth:** Bolton, England
**InfoReasoner's Reasoning Process:**
`<think>` To answer this question, I need to recall the band behind the album "Love Bites" and its formation city. Let's start with identifying the band.
`</think>`
`<search>` band Love Bites album
`</search>`
`<information>` Love Bites is the second studio album by English punk rock band Buzzcocks, released in 1978...
`</information>`
`<think>` Based on the search results, the correct band behind the album "Love Bites" is the English punk rock band Buzzcocks. Now, to find the formation city of the band Buzzcocks.
`</think>`
`<search>` formation city Buzzcocks band
`</search>`
`<information>` Buzzcocks are an English punk rock band, formed in Bolton, England, in 1976 by Pete Shelley and Howard Devoto...
`</information>`
`<think>` From the search results, the band Buzzcocks was formed in Bolton, England. Therefore, the album "Love Bites" was formed in the city of Bolton, England.
`</think>`
`<answer>` Bolton, England `</answer>`

*Figure 3.* Case study of InfoReasoner.

patterns. After this initial exploration phase, InfoReasoner's learning curve accelerates and eventually surpasses Search-R1, achieving a higher final EM score. This validates that the information gain component enables beneficial exploration, leading to a more robust retrieval policy and superior question-answering performance.

**Reward Component Analysis.** Figure 2b decomposes the total reward into the outcome-based EM score and the information gain (IG) score. The IG score provides a consistent dense reward signal, complementing the sparse EM supervision, while the EM score steadily increases throughout training, demonstrating that optimizing information gain translates into improved question-answering performance.

**Entropy Loss Analysis.** Figure 2c compares the entropy loss trajectories of InfoReasoner and Search-R1. InfoReasoner starts with a significant increase in entropy loss, reaching a peak higher than Search-R1, indicating that the

*Table 4.* Tool-integrated reasoning results on MATH500.

| Method | MATH500 |
|---|---|
| ZeroTIR-7B (Mai et al., 2025) | 0.802 |
| SimpleTIR-7B (Xue et al., 2025) | 0.846 |
| InfoReasoner-7B | **0.858** |

information gain reward encourages exploration of diverse retrieval queries and reasoning paths. Following this exploration phase, the entropy loss decreases sharply and stabilizes at a lower level, suggesting convergence to a more certain and confident policy.

**Response Length Analysis.** Figure 2d compares the response length trajectories of InfoReasoner and Search-R1 during training. InfoReasoner demonstrates a shorter and more stable response length, approximately 30% shorter than Search-R1. This indicates that InfoReasoner learns to generate more concise answers, as the information gain reward helps the model identify and prioritize key information from retrieved documents without sacrificing accuracy.

**Summary.** Together, these training dynamics provide mechanistic insights into why InfoReasoner achieves superior performance: it enables sufficient exploration (via information gain) while maintaining focus on task performance (via EM reward), striking the right balance between epistemic exploration and pragmatic exploitation. Furthermore, the shorter response length demonstrates that InfoReasoner not only achieves better accuracy but also generates more efficient outputs, highlighting the practical benefits of our approach.

### 4.5. Study on the Information Gain Reward

Figure 3 presents a case study demonstrating InfoReasoner's reasoning process on a multi-hop question, showing how it decomposes the query, conducts iterative searches, and synthesizes information to arrive at the correct answer.

Figure 4a presents the information gain values computed for different retrieval scenarios: providing only the first retrieved document (identifying Buzzcocks as the band behind "Love Bites"), providing only the second retrieved document (identifying Bolton as Buzzcocks' formation city), and providing both documents together. The boxplot reveals three patterns. First, *Info B* yields higher median information gain than *Info A*, suggesting the second document is more directly useful. Second, the naive *Sum* (Info A + Info B) im-

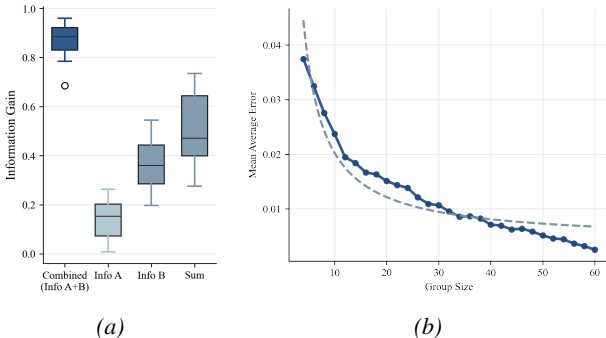

*Figure 4.* Information gain analysis: (a) Comparison across different retrieval scenarios demonstrating synergistic effects when jointly observing multiple documents; (b) Sensitivity analysis of group size $M$ on estimation accuracy, showing the trade-off between computational efficiency and reward signal quality.

proves over either document alone but falls below *Combined (Info A+B)*, indicating that jointly observing both documents reduces uncertainty more effectively than treating their contributions as additive. Third, *Combined* exhibits the highest central tendency, demonstrating a synergistic effect where the two documents together provide substantially more information than the sum of their individual contributions. This non-additive information gain validates our reward design that *assigns higher rewards to retrieval strategies gathering complementary evidence across reasoning hops*.

### 4.6. Sensitivity Analysis of Group Size on Reward Estimation

We investigate the trade-off between estimation accuracy and computational efficiency by varying the group size $M$. We establish a pseudo-ground truth oracle using $N = 64$ sequences and evaluate the estimation error of our information gain metric via bootstrap subsampling with $M \in \{4, \ldots, 60\}$. Figure 4b shows that the estimation error decays rapidly as $M$ increases, following the theoretical Monte Carlo convergence rate. The curve exhibits an elbow around $M = 12$, achieving low MAE ($< 0.02$) while reducing computational overhead by $4\times$ compared to the oracle baseline. Increasing $M$ beyond 12 yields diminishing returns. We adopt $M = 12$ as the optimal configuration, balancing reward signal quality and training efficiency.

## 5. Conclusion

We introduced *InfoReasoner*, a framework for optimizing agentic reasoning with retrieval through a synthetic semantic information gain reward. Theoretically, we redefine information gain as uncertainty reduction over belief states, establishing formal guarantees. Practically, we propose an output-aware semantic information gain reward that computes information gain from semantic equivalence classes using bidirectional textual entailment, eliminating the need for manual intermediate retrieval annotations. Experiments across seven QA benchmarks demonstrate that InfoReasoner achieves state-of-the-art performance at both 3B and 7B scales. The information gain component enables beneficial exploration during training, leading to more robust retrieval policies. This work provides a theoretically grounded and scalable path toward optimizing agentic reasoning with retrieval.

## Acknowledgements

This work was supported in part by the Hong Kong Innovation and Technology Commission under InnoHK Project CIMDA, in part by the Hong Kong SAR Government under the Global STEM Professorship and Research Talent Hub, and in part by the Hong Kong Jockey Club under the Hong Kong JC STEM Lab of Smart City (Ref.: 2023-0108).

## Impact Statement

The development of InfoReasoner has broader implications for the deployment of Agentic Reasoning Models in real-world scenarios:

*Advancing Trustworthy Agentic Search.* By grounding the model's motivation in "uncertainty reduction," we move closer to AI agents that actively verify their knowledge rather than hallucinating answers. This is critical for high-stakes applications like medical or legal information seeking, where the reliability of the reasoning process is as important as the final answer.

*Efficient Information Retrieval.* Unnecessary searching wastes computational resources and bandwidth. Our Information Gain theoretic validation encourages the model to search only when it lacks confidence, leading to more resource-efficient system designs that minimize API calls to external search engines.

*Generalizability of Semantic Metrics.* The proposed framework for estimating semantic uncertainty and information gain is not limited to QA. It provides a generalized methodology for measuring the "value of information" in any text-generation task, potentially benefiting fields such as automated scientific research, complex planning, and autonomous code generation.

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

# Appendix

## Content

## A. Related Work

### A.1. Agentic Reasoning

Large reasoning models (LRMs) aim to improve LLMs' test-time performance by incorporating extended reasoning steps. This approach differs from conventional large pre-trained models, which primarily scale during training by increasing model size or expanding the training dataset (Guo et al., 2025; Zhong et al., 2025). Recent research indicates that scaling models at test time can enhance the reasoning capabilities of smaller models when tackling complex tasks. Notably, models such as DeepSeek-R1 (Guo et al., 2025) and OpenAI o1 (Zhong et al., 2025) have been shown to explicitly employ chain-of-thought (CoT) reasoning (Wei et al.), effectively emulating human-like problem-solving strategies in areas like mathematics and coding.

In addition, some works start to integrate retrieval-augmented generation (RAG) and tools into LRMs' reasoning process, which means that the LRMs can retrieve external knowledge (knowledge base, web data, etc.) or tools such as calculators to assist in reasoning, thereby significantly enhancing the reasoning capabilities of the model. For example, Li *et al.* (Li et al., 2025a) introduced Search-o1, a framework that augments LRMs with an agentic RAG mechanism and a Reason-in-Documents module to refine retrieved content. This approach incorporates an agentic search workflow into the reasoning process, allowing LRMs to dynamically retrieve external knowledge when faced with uncertain information. Jin *et al.* (Jin et al., 2025a) introduced Search-R1, enabling a LRM to autonomously formulate multiple search queries throughout its step-by-step reasoning process, leveraging real-time retrieval to support each reasoning step. Furthermore, Xiong *et al.* (Xiong et al., 2025) presented RAG-Gym, a unified framework that systematically optimizes agentic reasoning by combining sophisticated prompt engineering, actor adjustment, and critic training. These works show that integrating RAG and tools into LRMs' reasoning process can significantly enhance the reasoning and complex problem-solving capabilities.

### A.2. Retrieval-Augmented Generation

Retrieval-Augmented Generation (RAG) has emerged as a promising paradigm to address the limitations of LLMs in factual consistency, knowledge coverage, and reasoning. It addresses the knowledge cut-off problem through a three-stage process (Li et al., 2025b). The first stage, *retrieval*, involves the LLM gathering relevant information from external sources (Xu et al.,

2025; Zhang et al., 2025a). In the second stage, *integration*, the retrieved content is deduplicated, conflicts are resolved, and the information is re-ranked for relevance (Zhao et al., 2024; Cheng et al., 2025). Finally, in the *generation* stage, the LLM reasons over the curated context to generate the final answer. For retrieval optimization, Xu *et al.* (Xu et al., 2025) proposed Collab-RAG to decompose complex queries to multiple simpler sub-queries, thereby enhancing the accuracy of the retrieval and facilitating more effective reasoning. Zhang *et al.* (Zhang et al., 2025a) introduced the PAR-RAG framework, which improves multi-step reasoning by selecting exemplars whose semantic complexity aligns with the current question, thereby enabling complexity-aware, top-down planning. For integration stage enhancement, there are two main approaches, relevance assessment and information synthesis. For example, Zhao *et al.* (Zhao et al., 2024) proposed SEER to extract evidence from retrieved passages to reduce computational costs and enhance the final RAG performance, and Cheng *et al.* (Cheng et al., 2025) proposed a dual-process approach that integrates reasoning-augmented querying with progressive knowledge aggregation, enabling the filtering and structuring of retrieved information into a continuously evolving outline. For generation stage enhancement, context-aware synthesis and grounded generation control are leveraged. For example, Islam *et al.* (Islam et al., 2024) proposed OpenRAG to dynamically select knowledge modules to ensure outputs remain relevant while reducing noise, while AlignRAG (Wei et al., 2026) leveraged critique-guided alignment to refine the generated path. However, our method is different from these methods in that we propose a synthetic semantic information gain reward to end-to-end guide the RAG and reasoning process.

### A.3. Reinforcement Learning for LLM Reasoning

Reinforcement learning (RL) has become an influential approach for improving the reasoning capabilities of LLMs. The application of RL to LLMs began with Reinforcement Learning from Human Feedback (RLHF) (Ouyang et al., 2022), which fine-tunes models to better reflect human preferences by leveraging feedback from human annotators. Foundational methods such as Proximal Policy Optimization (PPO) (Schulman et al., 2017) introduced robust policy optimization techniques using clipped objectives and reward normalization. More recently, methods like Direct Preference Optimization (DPO) (Rafailov et al.) have further streamlined the alignment process by directly optimizing on preference data, eliminating the need for explicit reward modeling. Group Relative Policy Optimization (GRPO) (Shao et al., 2024) marks a notable step forward in enhancing reasoning within RL frameworks by overcoming several shortcomings of earlier methods. Unlike traditional approaches that require a value function, GRPO estimates baselines using group scores, which greatly reduces the computational resources needed for training. This technique has been successfully applied in models such as DeepSeekMath (Shao et al., 2024) and DeepSeek-R1 (Guo et al., 2025), leading to improved mathematical reasoning performance. Moreover, GRPO and its derivatives have seen extensive application in agentic reasoning, where they are employed to optimize tool utilization, RAG, and reasoning trajectories, thereby further boosting the reasoning performance of LLMs. Notable examples include their use in Search-o1 (Li et al., 2025a), Search-R1 (Jin et al., 2025a), RAG-Gym (Xiong et al., 2025), and RAG-R1 (Tan et al., 2025). Recent concurrent work also explores information-theoretic or semantic-uncertainty objectives for reasoning optimization. InfoReasoner differs from IGPO (Wang et al., 2025) in that our reward is computed in semantic class space and measures the retrieval-conditioned increase in probability assigned to the gold semantic class, rather than directly rewarding answer-probability changes at the surface form level. It also differs from EMPO (Zhang et al., 2025b), which optimizes general reasoning through unsupervised semantic entropy minimization. InfoReasoner is retrieval-specific and uses a gold-class reward to provide dense credit for intermediate retrieval actions while remaining aligned with final-answer correctness.

## B. Theoretical Derivations and Proofs

### B.1. Derivation of LLM Belief Update

Here we provide the detailed derivation of the belief update rule for LLM agents, justifying the simplification used in Section 2.

In the context of LLM-based reasoning agents, the belief update must account for the fact that observations are generated through a two-stage process involving both environmental retrieval and LLM generation. The agent takes action $a_t$ (e.g., retrieval), and the environment returns evidence $e_t \sim P(\cdot \mid Y = y, a_t)$. The LLM then generates an observation $O_t \sim P_{\text{LLM}}(\cdot \mid e_t, b_t)$.

The complete observation likelihood is given by marginalizing over the retrieved evidence $e_t$:

$$P(O_t \mid Y = y, a_t, b_t) = \mathbb{E}_{e_t \sim P(\cdot \mid Y=y,a_t)} \left[ P_{\text{LLM}}(O_t \mid e_t, b_t) \right]. \tag{19}$$

The belief state is then updated according to the generalized Bayes rule:

$$b_{t+1}(y) = \frac{P(O_t \mid Y = y, a_t, b_t)\, b_t(y)}{\sum_{y' \in \mathcal{Y}} P(O_t \mid Y = y', a_t, b_t)\, b_t(y')}. \tag{20}$$

This formulation differs from standard POMDPs because the observation likelihood explicitly depends on the current belief $b_t$, reflecting how LLMs interpret and generate content based on their current understanding.

To make this tractable, we assume the LLM's interpretation of new evidence dominates its prior bias (i.e., $P_{\text{LLM}}(O_t \mid e_t, b_t) \approx P_{\text{LLM}}(O_t \mid e_t)$). Under this assumption, the likelihood simplifies to:

$$P(O_t \mid Y = y, a_t, b_t) \approx \mathbb{E}_{e_t \sim P(\cdot \mid Y = y, a_t)}[P_{\text{LLM}}(O_t \mid e_t)] = P(O_t \mid Y = y, a_t). \tag{21}$$

Substituting this back into Eq. (20) yields the standard form used in our analysis:

$$b_{t+1}(y) = \frac{P(O_t \mid Y = y, a_t)\, b_t(y)}{\sum_{y' \in \mathcal{Y}} P(O_t \mid Y = y', a_t)\, b_t(y')}. \tag{22}$$

**Discussion on Simplification Assumptions.** We acknowledge that the conditional independence assumption ($P_{\text{LLM}}(O_t \mid e_t, b_t) \approx P_{\text{LLM}}(O_t \mid e_t)$) represents an idealization where the LLM fully grounds its generation in the retrieved evidence, effectively suppressing conflicting prior biases. In practice, LLMs may exhibit "confirmation bias" or specific parametric knowledge tailored to the query, where the prior belief $b_t$ continues to influence generation despite new evidence $e_t$. The divergence from this assumption means that our theoretical bounds on uncertainty reduction effectively serve as an upper bound on performance in ideal grounding scenarios. However, this simplification is necessary to construct a tractable reward signal, and empirically, minimizing the uncertainty under this model actively encourages the policy to approximate this ideal behavior by seeking high-quality evidence that overwhelms prior ambiguity.

### B.2. Proof of Proposition 2.6 (Non-negativity under Ideal Updates)

This proof establishes that under ideal Bayesian updates, information gathering is never detrimental to the agent's knowledge state in expectation.

*Proof.* The non-negativity of Expected Information Gain (EIG) follows directly from the *Expected Non-Increase* axiom of the uncertainty functional $\mathcal{U}$. By Eq. (6), we have:

$$\mathbb{E}_{O \sim P(\cdot \mid b, a)}[\mathcal{U}(b_{t+1})] \leq \mathcal{U}(b). \tag{23}$$

Subtracting the expected posterior uncertainty from the prior uncertainty $\mathcal{U}(b)$, we obtain:

$$\text{EIG}(a \mid b) = \mathcal{U}(b) - \mathbb{E}_{O \sim P(\cdot \mid b, a)}[\mathcal{U}(b_{t+1})] \geq 0. \tag{24}$$

Equality holds (EIG = 0) if and only if the posterior belief equals the prior belief almost surely for all possible observations $O$. This occurs when the observation carries no information about the latent task variable $Y$, i.e., $O \perp Y \mid (b, a)$.

In the special case where $\mathcal{U}$ is the Shannon entropy, $\mathcal{U}_H(b) = H(b)$, the EIG coincides with the conditional mutual information $I(Y; O \mid a, b)$. Since mutual information is always non-negative, the *expected non-increase* property is naturally satisfied without requiring it as an independent axiom. $\square$

This result motivates maximizing EIG as an ideal objective under consistent Bayesian updates. In practice, we optimize the realized signal $\text{IG}_t$, which may be negative and thus naturally penalizes misleading retrievals.

### B.3. Proof of Proposition 2.7 (Telescoping Additivity)

This proof demonstrates that the sum of step-wise information gains is mathematically equivalent to the total reduction in uncertainty over a complete reasoning trajectory.

*Proof.* Consider a sequence of belief states $b_0, b_1, \ldots, b_T$ generated along a trajectory. By Definition 2.4, the realized information gain at each step $t$ is:

$$\text{IG}_t = \mathcal{U}(b_t) - \mathcal{U}(b_{t+1}). \tag{25}$$

Summing these incremental gains over the entire trajectory from $t = 0$ to $t = T - 1$:

$$\sum_{t=0}^{T-1} \text{IG}_t = \sum_{t=0}^{T-1} (\mathcal{U}(b_t) - \mathcal{U}(b_{t+1})) \tag{26}$$
$$= (\mathcal{U}(b_0) - \mathcal{U}(b_1)) + (\mathcal{U}(b_1) - \mathcal{U}(b_2)) + \cdots + (\mathcal{U}(b_{T-1}) - \mathcal{U}(b_T)).$$

Notice that the intermediate terms $\mathcal{U}(b_1), \ldots, \mathcal{U}(b_{T-1})$ cancel out, leaving only the initial and final uncertainty values:

$$\sum_{t=0}^{T-1} \text{IG}_t = \mathcal{U}(b_0) - \mathcal{U}(b_T). \tag{27}$$

$\square$

Telescoping additivity is a critical property for reinforcement learning. It proves that by maximizing the immediate, dense reward $\text{IG}_t$ at each step, the agent is effectively optimizing the global objective of minimizing final uncertainty about the answer.

### B.4. Proof of Proposition 2.8 (Monotonicity)

This proof ensures that our framework rationally prefers more informative sources (channels) over less informative ones, consistent with the theory of Blackwell dominance.

*Proof.* Let $a_1$ and $a_2$ be two information-gathering actions inducing observation channels. If $a_1$ Blackwell-dominates $a_2$, there exists a stochastic kernel $P(O_2 \mid O_1)$ such that the observation $O_2$ can be simulated by post-processing $O_1$.

When $\mathcal{U}$ is an $f$-divergence-based functional like Shannon entropy, the expected information gain is equivalent to the mutual information $I(Y; O \mid a, b)$. By the *Data Processing Inequality* for mutual information:

$$I(Y; O_1 \mid a_1, b) \geq I(Y; O_2 \mid a_2, b), \tag{28}$$

which directly implies:

$$\text{EIG}(a_1 \mid b) \geq \text{EIG}(a_2 \mid b), \quad \forall b. \tag{29}$$

For general uncertainty functionals satisfying the axioms, this monotonicity holds because any information reachable via $a_2$ is already contained (potentially with less noise) within the channel of $a_1$. $\square$

This property guarantees that the agent will systematically favor higher-quality information sources, such as more reliable knowledge bases or more precise reasoning tools, when multiple options are available.

## C. Algorithm Details

### C.1. Information Gain Reward Computation

Algorithm 1 details the step-by-step procedure for computing the information gain (IG) reward, which serves as dense intermediate supervision for the policy to seek informative evidence. The core challenge is estimating the model's epistemic uncertainty during the reasoning process.

We address this by measuring the reduction in semantic entropy before and after incorporating new evidence. As shown in Steps 1 and 2, the model samples a group of $M$ candidate responses for the current reasoning path. These responses are then clustered into semantic meaning classes $\{c_k\}$ to filter out surface-level linguistic variations and focus on core semantic uncertainty. Here we leverage DeBERTa-large model (He et al., 2021) that is fine-tuned on the NLI dataset MNLI (Williams et al., 2018) as NLI model $P_{\text{NLI}}$. By computing the entropy of the resulting belief distribution, we quantify how confused the model is about the final answer. The information gain is then defined as the difference between the prior and posterior

---

**Algorithm 1** Information Gain Reward Computation

---

1: **Input:** Question $x$, retrieved evidence $e_t$, reasoning model $\pi_\theta$, group size $M$, golden answer $y^\star$, NLI model $P_{\text{NLI}}$, threshold $\tau$.
2: **Output:** Information gain reward $\widehat{\text{IG}}_t$.
3: **Function** ESTIMATEBELIEF($Z, y^\star$):
4:     // 1. Sample semantic variation from the model
5:     Sample $M$ candidate responses $\mathcal{S} = \{s_1, \ldots, s_M\} \sim \pi_\theta(\cdot \mid Z)$.
6:     Compute sequence likelihoods $\{p_\theta(s_i \mid Z)\}_{i=1}^{M}$.
7:     // 2. Construct Semantic Equivalence Graph via Bidirectional Entailment
8:     Initialize undirected graph $G = (V, E)$ with nodes $V = \{1, \ldots, M\}$.
9:     **for** $i = 1$ **to** $M$, $j = i + 1$ **to** $M$ **do**
10:         $p_{\text{fwd}} \leftarrow P_{\text{NLI}}(s_i \vDash s_j \mid x)$;    $p_{\text{bwd}} \leftarrow P_{\text{NLI}}(s_j \vDash s_i \mid x)$.
11:         **if** $p_{\text{fwd}} > \tau$ **and** $p_{\text{bwd}} > \tau$ **then**
12:             Add edge $(i, j)$ to $E$. // Connect semantically equivalent answers
13:         **end if**
14:     **end for**
15:     // 3. Estimate Belief Distribution over Semantic Classes
16:     Extract connected components as semantic classes $\mathcal{C} = \{c_1, \ldots, c_K\}$.
17:     **for** each class $c_k \in \mathcal{C}$ **do**
18:         $p(c_k \mid Z) \leftarrow \sum_{s \in c_k} p_\theta(s \mid Z)$.
19:     **end for**
20:     // 4. Identify Ground-Truth Consistent Class
21:     Find correct class $c^\star \in \mathcal{C}$ such that $\exists s \in c^\star, s \leftrightarrow y^\star$ (via bi-NLI).
22:     **return** $p(c^\star \mid Z)$.
23: **End Function**
24: // Phase 1: Estimate Prior Belief (Conditioned on Question only)
25: Construct prior context $B(x) \leftarrow$ fact-free paraphrase of $x$.
26: $p(c^\star \mid B(x)) \leftarrow$ ESTIMATEBELIEF($B(x), y^\star$).
27: // Phase 2: Estimate Posterior Belief (Conditioned on Question + Evidence)
28: Construct posterior context $C_t(x) \leftarrow x \oplus e_t$.
29: $p(c^\star \mid C_t(x)) \leftarrow$ ESTIMATEBELIEF($C_t(x), y^\star$).
30: // Phase 3: Compute Information Gain as Uncertainty Reduction
31: $\widehat{\text{IG}}_t \leftarrow \log p(c^\star \mid C_t(x)) - \log p(c^\star \mid B(x))$.
32: **return** $\widehat{\text{IG}}_t$.

---

entropies. A positive value indicates that the retrieved evidence successfully concentrated the model's belief on a specific answer, thereby rewarding the informativeness of the retrieval action.

As discussed in Section D, we adopt $M = 12$ to strike a balance between estimation variance and computational efficiency. In practice, Step 1 (prior entropy) can often be pre-calculated or reused from the previous step's posterior entropy to further reduce overhead.

### C.2. InfoReasoner Training Algorithm

Algorithm 2 summarizes the complete training and reasoning architecture of InfoReasoner. The framework integrates an agentic iterative retrieval loop with the Group Relative Policy Optimization (GRPO) algorithm to optimize the retrieval and reasoning policy.

The training process follows a "sample-then-optimize" paradigm. For each question, we sample a group of $G$ independent trajectories (trajectories are sampled in parallel to enable the group-relative advantage estimation characteristic of GRPO). Each trajectory consists of multiple "Search-and-Reason" turns, where the model decides whether to generate a search query or proceed to the final answer based on its current state. During each turn, if a search query is generated, the environment returns the top-$k$ relevant documents, and the model receives an intermediate IG reward (computed via Algorithm 1) that quantifies the value of the newly acquired information.

---

**Algorithm 2** InfoReasoner Training Algorithm

---

1: **Input:** Dataset $\mathcal{D}$, initial policy $\pi_\theta$, reference policy $\pi_{\text{ref}}$, max turns $T$, group size $G$.
2: **for** each training iteration **do**
3:     Sample a question $x$ and ground truth $y^\star$ from $\mathcal{D}$.
4:     // Phase 1: Group Rollout (Sample $G$ diverse trajectories)
5:     **for** $i = 1$ to $G$ **do**
6:         Initialize state $s_{0,i} = \{x\}$.
7:         Initialize trajectory history $\mathcal{H}_i = \emptyset$.
8:         **for** $t = 1$ to $T$ **do**
9:             Generate thought $h_{t,i}$ and action $a_{t,i} \sim \pi_\theta(\cdot \mid s_{t-1,i})$.
10:            **if** $a_{t,i}$ is `<search>` query **then**
11:                Retrieve top-$k$ documents $e_{t,i} \leftarrow \text{Env}(a_{t,i})$.
12:                Update state $s_{t,i} \leftarrow s_{t-1,i} \oplus h_{t,i} \oplus a_{t,i} \oplus e_{t,i}$.
13:                Record step $\mathcal{H}_i \leftarrow \mathcal{H}_i \cup \{(x, a_{t,i}, e_{t,i})\}$.
14:            **else if** $a_{t,i}$ is `<answer>` **then**
15:                Extract predicted answer $\hat{y}_i$.
16:                Record terminal answer $\hat{y}_i$.
17:                **break**
18:            **end if**
19:         **end for**
20:         // Phase 2: Post-hoc Reward Computation
21:         Compute outcome reward $r^{\text{out}} \leftarrow \mathbb{I}(\hat{y}_i = y^\star)$.
22:         Initialize traversal rewards $\mathcal{R}_i^{\text{int}} = \emptyset$.
23:         **for** each step $(x, a, e) \in \mathcal{H}_i$ **do**
24:            Compute intrinsic reward $r \leftarrow$ Algorithm $1(x, e, \dots)$.
25:            $\mathcal{R}_i^{\text{int}} \leftarrow \mathcal{R}_i^{\text{int}} \cup \{r\}$.
26:         **end for**
27:         Calculate total reward: $R_i = r^{\text{out}} + \lambda \cdot \frac{1}{|\mathcal{R}_i^{\text{int}}|} \sum_{r \in \mathcal{R}_i^{\text{int}}} r$.
28:     **end for**
29:     // Phase 3: Group Relative Policy Optimization
30:     Compute group statistics: $\mu_R = \frac{1}{G} \sum R_i$, $\sigma_R = \sqrt{\frac{1}{G} \sum (R_i - \mu_R)^2}$.
31:     Compute advantages: $A_i = \frac{R_i - \mu_R}{\sigma_R + \epsilon}$.
32:     Update $\pi_\theta$ to maximize GRPO objective:
33:     $\mathcal{L}(\theta) = \frac{1}{G} \sum_{i=1}^{G} \left[ \min \left( \frac{\pi_\theta(o_i|x)}{\pi_{\text{old}}(o_i|x)} A_i, \text{clip} \left( \frac{\pi_\theta(o_i|x)}{\pi_{\text{old}}(o_i|x)}, 1 - \epsilon, 1 + \epsilon \right) A_i \right) - \beta \mathbb{D}_{\text{KL}}(\pi_\theta \| \pi_{\text{ref}}) \right]$.
34: **end for**

---

After completing the reasoning process (either by reaching a terminal state or the maximum turn limit $T$), we compute a composite reward $R_i$ for each trajectory. This reward combines the sparse outcome-based Exact Match (EM) score with the cumulative intrinsic IG rewards, balanced by the coefficient $\lambda$. Finally, the policy is updated by comparing each trajectory's reward against the group mean, effectively encouraging the model to favor retrieval strategies that are more "efficient" and "informative" than its current average performance.

## D. Implementation Details

### D.1. Training Details

InfoReasoner was trained on 8 NVIDIA A800 GPUs using full-parameter fine-tuning. The training dataset was created by merging NQ and HotpotQA, and this same dataset was used for both InfoReasoner and all training-based baselines. Distributed training was conducted with Fully Sharded Data Parallelism (FSDP), and BFloat16 precision was employed throughout both training and evaluation. We leverage VeRL (Sheng et al., 2025) framework for training and the hyper-parameters are set as Table 5. The main QA experiments use Qwen2.5-3B and Qwen2.5-7B backbones. The MATH500 tool-integrated experiment uses Qwen2.5-7B, and the WebDetective experiment uses the 3B-scale InfoReasoner policy. For

*Table 5.* Key hyperparameters used for InfoReasoner training. Unless otherwise specified, these apply to both 3B and 7B models.

| Hyperparameter | Value |
|---|---|
| Training Batch Size | 512 |
| Testing Batch Size | 256 |
| Max Observation Length | 500 |
| Micro Training Batch Size | 64 (3B), 32 (7B) |
| Actor Learning Rate | $1 \times 10^{-6}$ |
| Training Steps | 300 |
| Number of Maximum Search Turns (Training) | 2 |
| KL Loss Coefficient | 0.001 |
| Rollout Group Size | 3 |
| Maximum Retrieval Documents | 3 |
| Tensor Model Parallel Size | 1 (3B), 4 (7B) |
| Group Size on Reward Estimation | 12 |
| Information Gain Coefficient | 0.6 |

---

**System Prompt**

Answer the given question. You must conduct reasoning inside `<think>` and `</think>` first every time you get new information. After reasoning, if you find you lack some knowledge, you can call a search engine by `<search>` query `</search>`, and it will return the top searched results between `<information>` and `</information>`. You can search as many times as you want. If you find no further external knowledge needed, you can directly provide the answer inside `<answer>` and `</answer>` without detailed illustrations. For example, `<answer>` xxx `</answer>`. Question: {question}.

*Figure 5.* System prompt used for training and inference, guiding the model to perform iterative reasoning and retrieval.

efficient retrieval, we deploy a local search engine on 4 NVIDIA A800 GPUs. We use the E5 encoder (Wang et al., 2024) and the Wikipedia dump from FlashRAG (Jin et al., 2025b) as the retrieval backend for open-domain and multi-hop QA tasks. For complex, real-time web-based tasks, we cache Google Search results and scrape the full content of retrieved pages. Retrieval over this corpus is performed using the BGE-M3 retriever (Chen et al., 2024).

### D.2. Prompt Details

In this section, we provide the specific prompts used throughout the training and evaluation of InfoReasoner. These prompts are designed to maintain structural consistency and ensure the model effectively leverages the agentic search capabilities.

*System Prompt.* The prompt shown in Figure 5 is the primary instruction provided to the model during both the RL training and inference phases. It guides the model to conduct iterative reasoning within `<think>` tags and execute retrieval actions using `<search>` tags.

*Error Handling Prompt.* To maintain the stability of the agentic loop, we employ a dedicated error handling prompt. When the model generates a response that does not follow the predefined `<search>` or `<answer>` formats, this prompt is appended to the context to correct the model and re-induce the valid action format. The prompt is shown in Figure 6.

*Information Gain Estimation Prompts.* To estimate the belief distribution for information gain computation, we sample

*Table 6.* Inference efficiency comparison: number of search turns required to match or surpass the peak performance of the Search-R1 baseline.

| Dataset | Method | EM @ $T = 2$ | Baseline Peak EM | Turns to Match Peak |
|---|---|---|---|---|
| HotpotQA | Search-R1 | 32.4% | 34.6% ($T = 6$) | 6.0 |
| | **InfoReasoner** | **41.4%** | - | **2.0 (↓ 66.7%)** |
| 2Wiki | Search-R1 | 19.4% | 30.1% ($T = 4$) | 4.0 |
| | **InfoReasoner** | **30.2%** | - | **2.0 (↓ 50.0%)** |

---

**Error Handling Prompt**

My previous action is invalid. If I want to search, I should put the query between `<search>` and `</search>`. If I want to give the final answer, I should put the answer between `<answer>` and `</answer>`. Let me try again.

---

*Figure 6.* Error handling prompt used to correct invalid model actions and guide proper format compliance.

---

**Prompts for Information Gain Estimation**

**Prompt for Naive Generation**
Answer the question based on your own knowledge. Only give me the answer and do not output any other words.
Question: {question}

**Prompt for Generation with Retrieved Documents**
Answer the question based on the given document. Only give me the answer and do not output any other words.
The following are given documents. {documents}
Question: {question}

---

*Figure 7.* Prompts used for sampling candidate responses to estimate semantic belief distributions for information gain computation.

candidate responses using specific prompts for both naive (prior) and retrieval-augmented (posterior) contexts. The prompts are shown in Figure 7.

# E. Additional Experimental Results

This section provides extended analysis and ablation studies that complement our main findings.

## E.1. Training and Inference Efficiency

In this section, we analyze the efficiency of InfoReasoner from both training and inference perspectives.

*Training efficiency.* While the computation of the Information Gain (IG) reward introduces additional complexity during training due to the need for multiple samples and NLI checks, this overhead is primarily restricted to the training phase. Specifically, estimating semantic entropy using $M = 12$ samples increases the training time per step compared to standard GRPO with pure outcome rewards. However, this is a one-time cost that yields a significantly more robust policy. Using the non-IG portion as a proxy for outcome-only training, semantic IG introduces a $1.9\times$ training slowdown over 300 steps. The implied extra cost is 18.82 wall-clock hours, or 150.53 GPU-hours, and is incurred only during RL training.

*Inference efficiency.* Crucially, at inference time, InfoReasoner does not require any additional sampling or NLI checks. As shown in Table 6, the learned policy leads to significantly more efficient information seeking. On HotpotQA, InfoReasoner surpasses the peak performance of the Search-R1 baseline (34.6%) after only 2 turns, representing a 66.7% reduction in retrieval calls. Similarly, on 2Wiki, it achieves peak-matching performance with 50.0% fewer turns. This efficiency stems from its ability to prioritize high-gain queries that resolve maximum uncertainty early in the reasoning process.

## E.2. Ablation Study on Search Turns

We study how the number of search turns affects performance by varying $T$ from 1 to 8. Figure 8 shows EM scores for InfoReasoner and Search-R1 on HotpotQA and 2WikiMultihopQA.

*Large gains from single-turn to two-turn retrieval.* Both methods improve substantially from $T = 1$ to $T = 2$. On HotpotQA, InfoReasoner improves from 11.0% to 41.4% EM (+30.4 points), while Search-R1 increases from 9.8% to 32.4% (+22.6 points). On 2WikiMultihopQA, InfoReasoner rises from 1.6% to 30.2% (+28.6 points), compared to Search-R1 from 1.6% to 19.4% (+17.8 points). This shows that multi-turn retrieval is essential for multi-hop reasoning.

*Diminishing returns beyond 3-4 search turns.* Performance gains diminish after $T = 2$. On HotpotQA, InfoReasoner plateaus at 42.6% by $T = 4$ (maintained at $T = 6$ and $T = 8$), while Search-R1 plateaus at 34.6% by $T = 6$. On 2WikiMultihopQA, both methods plateau by $T = 4$ (InfoReasoner: 39.5%; Search-R1: 30.1%). InfoReasoner's earlier

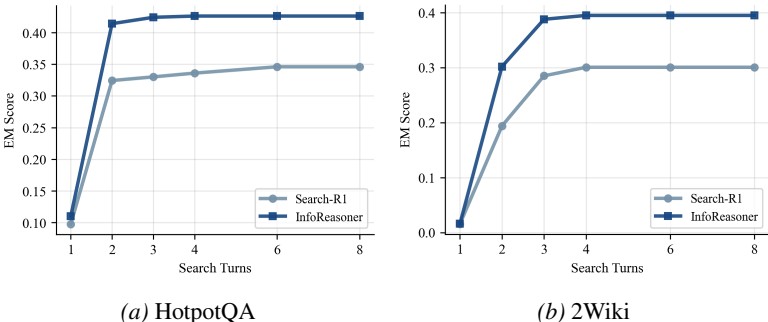

*(a)* HotpotQA           *(b)* 2Wiki

*Figure 8.* Search turns analysis for InfoReasoner and Search-R1 in HotpotQA and 2Wiki multi-hop QA datasets.

*Table 7.* Training-time cost breakdown for InfoReasoner on 8×A800 80GB GPUs.

| Quantity | Value |
|---|---|
| Number of steps | 300 |
| Avg total time / step | 472.2 s |
| Avg IG time / step | 225.8 s |
| Avg non-IG time / step | 246.4 s |
| Wall-clock | 39.35 h |
| GPU-hours | 314.8 |

saturation at higher performance demonstrates its superior information-gain efficiency per query.

*InfoReasoner outperforms Search-R1 across all settings.* At $T = 2$, InfoReasoner leads by +9.0 EM on HotpotQA (41.4% vs. 32.4%) and +10.8 EM on 2WikiMultihopQA (30.2% vs. 19.4%). This advantage comes from maximizing expected information gain: InfoReasoner prioritizes evidence that reduces semantic uncertainty most, avoiding redundant queries and achieving higher accuracy with fewer turns.

### E.3. More Case Studies

**Negative Information Gain as a Quality Signal.** Figure 9 demonstrates how our reward mechanism effectively handles misleading evidence. When the retriever surfaces information about a song instead of the target magazine, the framework assigns a *negative information gain reward* $(-0.915)$. This negative realized gain confirms that the estimator correctly detects the increase in epistemic uncertainty caused by irrelevant or "poisoned" context. Such cases validate that InfoReasoner provides a precise feedback signal, steering the policy away from deceptive reasoning paths and toward more reliable information-seeking strategies.

**Advantages of Information Gain Reward over Pure Outcome Reward.** Figure 10 highlights a key advantage of our information gain metric: its ability to provide dense, meaningful credit assignment even when the final task fails. Here, the retrieval action successfully surfaced highly relevant documents identifying Samuel L. Jackson as the hitman (Darius Kincaid). Our framework assigns a high positive IG (0.808) because the evidence significantly reduced semantic uncertainty and concentrated the model's belief on the correct class. In contrast, a standard outcome-based reward (EM score) would assign a zero reward to the entire trajectory due to the final reasoning error. By rewarding correctly-intentioned and effective information gathering independently of final reasoning performance, InfoReasoner decouples the learning of *retrieval strategies* from the learning of *information synthesis*. This ensures that the agent is not unfairly penalized for a good search action that happened to be followed by a reasoning slip, leading to more stable and efficient policy optimization.

## F. Limitations and Future Work

A primary limitation of our framework lies in the computational overhead associated with the Information Gain reward during training. Deriving an accurate reward signal requires estimating the semantic entropy of the model's outputs, which involves sampling multiple candidate sequences (e.g., $M = 12$) and performing pairwise NLI inferences to construct semantic equivalence classes. This process imposes a burden on training resources compared to standard outcome-based RL.

---

**Case Study: Misleading Retrieval**

**Query:** Are Instinct and Popular Mechanics both magazines?
**Ground-truth:** Yes
**Reasoning Process:**
`<think>` I need to determine if Instinct and Popular Mechanics are both magazines. I'll start by searching for information about Instinct and Popular Mechanics. `</think>`
`<search>` "Instinct" "Popular Mechanics" `</search>`
`<information>`
Doc 1 (Title: "Instinct (song)") Instinct (song) "Instinct" is a 1996 song by rock group Crowded House. It was the first single released from the group's greatest hits compilation "Recurring Dream" in the United Kingdom, and the third and final release in Australia. It was a top 20 hit in New Zealand and the UK, peaking at number 17 and number 12, respectively; in the UK, it became their second-highest peak alongside "Locked Out" (their highest peak was "Weather with You" at number seven). In Australia, "Instinct" peaked at number 90 on the ARIA Singles Chart in March 1997, spending two non-consecutive weeks in the ...
Doc 2 (Title: "Instinctive drift") automatic and unplanned, a natural reaction which often is preferred by the animal over learned and unnatural actions. This topic is relevant and discussed when it comes to the nature and nurture controversy. Many experimental studies have been conducted on this topic which proves its reliability. Natural behaviours are easier to learn and often easier expressed, even subliminally. The term instinctive drift was coined by Keller and Marian Breland, former students of B.F. Skinner at the University of Minnesota (Marian in 1938, Keller in 1940). B.F. Skinner coined the term operant conditioning. The Breland's followed Skinners principles of control and ...
Doc 3 (Title: "Popular Mechanics") Popular Mechanics Popular Mechanics is a magazine of popular science and technology. Popular Mechanics was first published by Henry Haven Windsor, January 11, 1902. It has been owned since 1958 by Hearst Communications. There are nine international editions, including a now-defunct Latin American version that had been published for decades, and a newer South African edition. The Russian edition of "Popular Mechanics" has been issued since 2002. In 2013, the US edition changed from twelve to ten issues per year. "Popular Mechanics" features regular sections on automotive, home, outdoors, science, and technology topics. A recurring column is "Jay Leno's Garage"
`</information>`
`<think>` After searching for information about Instinct and Popular Mechanics, it appears that Instinct is a song by Crowded House, while Popular Mechanics is a magazine of popular science and technology. Since they have different forms of media and content, Instinct and Popular Mechanics are not both magazines. `</think>`
`<answer>` No `</answer>`

**Information Gain Score:** -0.915

*Figure 9.* Case study on misleading retrieval. When the retriever surfaces irrelevant information (e.g., a song instead of the target magazine), InfoReasoner assigns a negative information gain reward (-0.915). This correctly identifies the increase in epistemic uncertainty and provides a vital corrective signal for policy optimization.

However, it is crucial to note that this overhead is strictly confined to the training phase. At inference time, InfoReasoner operates efficiently by simply executing the learned policy without any auxiliary sampling or NLI computation. Future work aims to mitigate the training cost by exploring lightweight approximations, such as estimating semantic uncertainty directly from the model's internal hidden states or replacing the expensive NLI model with calibrated embedding-based clustering methods. These advancements would further enhance the scalability of our approach for larger models and broader domains.

The development of InfoReasoner has broader implications for the deployment of Agentic Reasoning Models in real-world scenarios:

*Advancing Trustworthy Agentic Search.* By grounding the model's motivation in "uncertainty reduction," we move closer to AI agents that actively verify their knowledge rather than hallucinating answers. This is critical for high-stakes applications like medical or legal information seeking, where the reliability of the reasoning process is as important as the final answer.

*Efficient Information Retrieval.* Unnecessary searching wastes computational resources and bandwidth. Our Information Gain theoretic validation encourages the model to search only when it lacks confidence, leading to more resource-efficient system designs that minimize API calls to external search engines.

*Generalizability of Semantic Metrics.* The proposed framework for estimating semantic uncertainty and information gain is not limited to QA. It provides a generalized methodology for measuring the "value of information" in any text-generation task, potentially benefiting fields such as automated scientific research, complex planning, and autonomous code generation.

---

**Case Study: High Information Gain with Reasoning Failure**

**Query:** who plays the hitman in the hitman's bodyguard?
**Ground-truth:** Samuel L. Jackson
**Reasoning Process:**
`<think>` To find out who plays the role of the hitman in the movie "The Hitman's Bodyguard," I need to search for the cast of this movie. `</think>`
`<search>` who plays the hitman in the hitman's bodyguard `</search>`
`<information>`
Doc 1 (Title: "The Hitman's Bodyguard") The Hitman's Bodyguard The Hitman's Bodyguard is a 2017 American action thriller comedy film directed by Patrick Hughes and written by Tom O'Connor, whose script was on the 2011 Black List survey. The film stars in main roles Ryan Reynolds and Samuel L. Jackson which are side by side from Gary Oldman and Salma Hayek, and follows a bodyguard named Michael Bryce who is hired to protect a popular hitman named Darius Kincaid who has to testify at the International Criminal Court. "The Hitman's Bodyguard" was released in the United States on August 18, 2017, and grossed $176 million worldwide...
Doc 2 (Title: "The Hitman's Bodyguard") was announced that Jackson, Reynolds and Hayek were in early talks to reprise their roles for a sequel, titled "The Hitman's Wife's Bodyguard", with plans to begin filming later in the year. The Hitman's Bodyguard The Hitman's Bodyguard is a 2017 American action thriller comedy film directed by Patrick Hughes and written by Tom O'Connor, whose script was on the 2011 Black List survey. The film stars in main roles Ryan Reynolds and Samuel L. Jackson which are side by side from Gary Oldman and Salma Hayek, and follows a bodyguard named Michael Bryce who is hired to protect a ...
Doc 3 (Title: "The Hitman's Bodyguard") The movie ends with them dancing and then kissing in the midst of all the fighting and chaos taking place at the bar. In May 2011, David Ellison's Skydance Media acquired the action script "The Hitman's Bodyguard" written by Tom O'Connor. The script was among the top 2011 Black List of unproduced screenplays. While originally intended as a drama, the script underwent a "frantic" two-week rewrite to be remade into a comedy several weeks prior to filming. On November 4, 2015, it was announced that Ryan Reynolds, Samuel L. Jackson and Gary Oldman were cast in the film, which Jeff ...
`</information>`
`<think>` From the search results, I can see that the actor who plays the role of the hitman in "The Hitman's Bodyguard" is Gary Oldman. `</think>`
`<answer>` Gary Oldman `</answer>`

**Information Gain Score:** 0.808

*Figure 10.* Case study on high Information Gain despite reasoning failure. Even when the reasoning chain concludes incorrectly, the positive information gain reward (0.808) correctly identifies and rewards the successful discovery of ground-truth evidence. This illustrates how InfoReasoner provides effective credit assignment for retrieval, decoupled from the final reasoning outcome.

