# OpenReview forum: "Optimizing Agentic Reasoning with Retrieval via Synthetic Semantic Information Gain Reward"
_ICML.cc/2026/Conference — ICML 2026 regular_

### Official Review · Reviewer_zP4S · 2026-03-10

**Soundness:** 3
**Presentation:** 3
**Significance:** 2
**Originality:** 3
**Overall Recommendation:** 4
**Confidence:** 3

**Summary:**

This paper proposes InfoReasoner. This framework adds a synthetic semantic information gain reward to train agentic RAG methods. The main idea is to sample multiple answers with and without retrieved evidence, cluster them into semantic equivalence classes via bidirectional NLI, and measure the reduction in semantic entropy as an intrinsic reward. This IG reward supplements the standard EM reward in GRPO training. Experiments show good consistent gains over search R1 and other retrieval augmented baselines at 3B and 7B scales.

**Compliance With Llm Reviewing Policy:**

Affirmed.

**Key Questions For Authors:**

(1) How sensitive are the results to the NLI model?  (2) How much slower is training compared to Search-R1?

**Limitations:**

yes

**Strengths And Weaknesses:**

Strengths

(1) Credit assignment for intermediate retrieval steps is a real bottleneck in agentic RAG training.  The ablation in table 2 confirms the IG signal helps with a clean 4.7-point gain.

(2) The empirical results are solid overall. The method shows fairly consistent improvements across different setting and at two scales.

(3) The training dynamics in Fig. 2 are informative. The case studies (e.g. Fig. 9/10) are well chosen, which well show the reward tracks retrieval quality.

(4) The paper is generally easy to follow. The motivation is clear, the method section is organized in a good way.

Weaknesses

(1) The theoretical section is somewhat lighter than the presentation initially suggests. Prop. 2.6 follows from mutual information being non-negative; prop. 2.7 is a standard telescoping sum; prop. 2.8 restates the data processing inequality; which are common techniques.

(2) The evaluation is restricted to short-answer QA settings, where exact match is an appropriate metric and semantic equivalence is relatively easier to assess. It would be better if the paper included at least some evidence on longer form generation task.

(3) To me, the novelty seems to lie more in the particular combination of existing ideas than in a new method. IG as intrinsic reward has been widely explored; same to the case that using semantic entropy by output clustering.

---

> ### Author Rebuttal · Authors · 2026-03-29
>
> Dear Reviewer zP4S,
>
> We greatly appreciate your thoughtful feedback.
>
> **W1: Theoretical contribution and the role of standard techniques.**
>
> **Reply**: We agree that the individual proof techniques are standard. Our intention is not to claim new information-theoretic identities, rather, the contribution lies in **formulating agentic retrieval under a belief-state and uncertainty-reduction perspective and showing how these standard properties justify the reward design used in InfoReasoner**.
>
> In particular, the role of Proposition 2.6 is to motivate information gain (IG) as a principled retrieval objective under idealized Bayesian updates, while our practical framework further distinguishes this expected quantity from the realized IG used during training, which can be negative and thus serves as a useful penalty for misleading retrieval.
>
> In addition, Proposition 2.7 clarifies why local step-wise rewards align with the global objective of reducing uncertainty over the full reasoning trajectory, and Proposition 2.8 provides a formal preference ordering over information channels.
>
> **W2: Evaluation scope and evidence on longer-form tasks.**
>
> **Reply**: We provide an additional experiment on **Tool-Integrated Reasoning (TIR)** using MATH500, where the model produces multi-step solutions interleaved with Python code execution, which has longer steps than short-answer QA. We adapt the IG reward by estimating prior/posterior beliefs before/after interpreter execution:
>
> | Method | MATH500 |
> |---|---|
> | ZeroTIR-7B (arxiv:2505.07773) | 0.802 |
> | SimpleTIR-7B (arxiv:2509.02479) | 0.846 |
> | InfoReasoner-7B (ours) | **0.858** |
>
> InfoReasoner-7B outperforms both baselines, this provides meaningful evidence that the proposed reward is not limited to short-answer retrieval QA and can transfer to longer, tool-augmented reasoning trajectories.
>
> **W3: Novelty beyond the combination of existing ideas.**
>
> **Reply**: We agree that the paper should better clarify where the novelty lies. The novelty of InfoReasoner lies not in introducing information gain or semantic clustering as standalone concepts, but in how these ideas are reformulated and instantiated for an important problem: *dense credit assignment for intermediate retrieval actions in agentic reasoning*.
>
> **On the theoretical side**, our contribution is to cast retrieval optimization in a belief-state and uncertainty-reduction framework and use this perspective to justify step-level retrieval rewards. In particular, the framework clarifies the role of information gain as a retrieval objective, distinguishes expected IG from the realized IG used in training, and explains why local retrieval rewards can align with global uncertainty reduction over a full reasoning trajectory. **On the practical side**, we develop a computable reward instantiation by estimating semantic answer beliefs from sampled outputs via clustering, and integrate this reward into GRPO to optimize retrieval behavior in agentic reasoning.
>
> Importantly, turning these ingredients into a working reward for agentic retrieval is not a trivial plug-in combination: the reward must be instantiated in a way that avoids rewarding mere confidence, supports step-level credit assignment, and remains balanced with final-task correctness.
>
> **Q1: How sensitive are the results to the NLI model?**
>
> **Reply**: We recomputed IG scores using different NLI backbones, keeping generation fixed to isolate the NLI impact:
>
>   | NLI Model | Pearson | Sign agreement |
>   |---|---|---|
>   | RoBERTa-large-MNLI | 0.944 | 0.883 |
>   | DeBERTa-large-MNLI-Fever-ANLI | 0.862 | 0.883 |
>   | DistilBART-MNLI-12-3 | 0.853 | 0.850 |
>
> Pearson $\geq$ 0.85 and sign agreement $\geq$ 0.85 across all backbones indicate that the reward signal is reasonably robust to NLI model choice: both the relative ranking of IG scores and the helpful-vs.-harmful judgment remain broadly stable.
>
> **Q2: How much slower is training compared to Search-R1?**
>
> **Reply**: Using the non-IG portion as a rough proxy for outcome-only training, the wall-clock / GPU-hour comparison on 8×A800 80GB is:
>
> | Quantity | Search-R1-style method | InfoReasoner |
> |---|---|---|
> | Avg time / step | 246.4 s | 472.2 s |
> | Wall-clock for 300 steps | 20.53 h | 39.35 h |
> | GPU-hours for 300 steps | 164.27 | 314.8 |
> | Relative slowdown | 1.0× | 1.9× |
>
> The implied extra cost of semantic IG is therefore about 18.82 wall-clock hours / 150.53 GPU-hours over 300 steps. More broadly, this should be viewed as a **trade-off** between training and inference: the extra cost is incurred only during RL training, while no multi-sample reward estimation or NLI-based clustering is required at inference time. Appendix E.1 further shows that the learned policy can reduce inference-time search cost with 50-67% fewer retrieval turns.

---

> > ### Author Rebuttal · Reviewer_zP4S · 2026-04-01
> >
> > Thank you for the rebuttal; my concerns are addressed and I maintain my positive score.

---

### Official Review · Reviewer_2hVh · 2026-03-10

**Soundness:** 3
**Presentation:** 3
**Significance:** 3
**Originality:** 3
**Overall Recommendation:** 4
**Confidence:** 3

**Summary:**

This paper proposes InfoReasoner, a retrieval-aware RL framework for agentic reasoning. The authors propose a synthetic semantic information gain reward to score retrieval actions. The method clusters sampled answers into semantic classes, measures entropy reduction after retrieval, and adds this signal to GRPO training. Experiments on several QA benchmarks show gains over other retrieval and non-retrieval baselines.

**Compliance With Llm Reviewing Policy:**

Affirmed.

**Key Questions For Authors:**

All concerns have been mentioned in the previous section.

**Limitations:**

yes

**Strengths And Weaknesses:**

Strengths:
1. The main idea is clear. Reward retrieval by reducing uncertainty, not only by final EM.
2. The method is practical. It avoids manual retrieval annotations and estimates reward from model outputs.
3. The empirical section is reasonably broad. It covers seven QA datasets and both 3B and 7B settings.

Weaknesses:
1. There is an equation-level inconsistency. Eq. (15) defines information gain using entropy reduction, while Eq. (17) redefines it using log-probability of the gold semantic class. The paper does not cleanly separate these two rewards, or I may have missed it. I appreciate if authors could clarify this.
2. The method depends on gold answers during training to identify the correct semantic class. That weakens the claim of being fully "intrinsic". It is annotation-free for retrieval steps, I agree, but not label-free overall.
3. There is no clear error analysis of failures. I want to know when this reward helps and when it hurts, compared to Search -R1 for instance.
4. Connected components over bidirectional entailment can be noisy and threshold-sensitive. The paper does not study threshold robustness. I would like to see ablations on this as well. Also, what happens with models from other family or from Qwen3 family?
5. Writing shoud be improved, typos and wording issues exist. Examples: “Optimxization” in the abstract, “Since the environment state s ... may not directly observable”.
6. Authors should double-check and fix wrong citations:
- ReSearch: Learning to Reason with Search for LLMs via Reinforcement Learning --> NeurIPS 2025
- DualRAG: A Dual-Process Approach to Integrate Reasoning and Retrieval for Multi-Hop Question Answering --> ACL 2025
- DeBERTa: Decodingenhanced BERT with Disentangled Attention --> ICLR 2021

---

> ### Author Rebuttal · Authors · 2026-03-29
>
> Dear Reviewer 2hVh,
>
> We greatly appreciate your thoughtful feedback.
>
> **W1: Relationship between Eq. (15) (entropy-based IG) and Eq. (17) (gold-class IG)**
>
> **Reply**: Thank you for this observation. We acknowledge that both formulations use the same notation $ \widehat{\mathrm{IG}}\_t $ without explicitly marking the transition, which causes confusion. We clarify below.
>
> Eq. (15) defines a general information gain based on entropy reduction across all semantic classes: $\widehat{\mathrm{IG}}\_t =H_{\text{sem}}(B)-H_{\text{sem}}(C_t)$. This is the natural instantiation of the theoretical framework, but has a practical limitation: entropy reduction can occur even when the model concentrates mass on a *wrong* class, creating a reward-hacking risk. Eq. (17) instead anchors the signal to the *gold semantic class*: $\widehat{\mathrm{IG}}\_t = \log p(c^\star \mid C_t) - \log p(c^\star \mid B)$, measuring whether retrieval increases belief specifically in the correct answer. This resolves the reward-hacking issue and is the formulation actually used in training.
>
> In summary, Eq. (15) provides the theoretical uncertainty-reduction objective, while Eq. (17) is the practical training reward used in our method and motivated by this principle.
>
> **W2: Role of gold answers and the scope of the "intrinsic" claim.**
>
> **Reply**: We thank the reviewer for this comment. To clarify, our paper does not intend to claim that the overall training procedure is fully intrinsic or fully label-free. The method still uses final-answer supervision through $\mathbb{I}(y_i = y^\star)$, and Eq. (17) also depends on the ground-truth answer via the gold semantic class $c^\star$. What we mean is that the IG component provides intermediate supervision for retrieval steps without requiring manual annotations for those intermediate steps. We agree that this distinction could be stated more clearly, and we will revise the wording accordingly.
>
> **W3: When does the IG reward help vs. hurt?**
>
> **Reply**: The IG reward helps when retrieval provides useful intermediate evidence that pure outcome supervision cannot credit. This is reflected in the $\lambda$ ablation, where adding IG ($\lambda=0.6$) improves average accuracy by 4.7 points over the IG-free baseline ($\lambda=0.0$), and in Fig. 9 and 10, where IG rewards helpful retrieval and penalizes misleading retrieval even when EM alone would be insufficient.
>
> It can hurt when over-emphasized: increasing $\lambda$ from 0.6 to 1.0 reduces accuracy from 0.346 to 0.320, indicating that excessive reliance on IG can conflict with final-task correctness.
>
> **W4: Threshold sensitivity of semantic clustering and transfer to Qwen3.**
>
> **Reply**: We agree that threshold robustness should be reported. We therefore ran a threshold ablation over the bidirectional-entailment threshold $\tau$ using Qwen2.5-3B:
>
> | Threshold $\tau$ | EM Score |
> |---|---|
> | 0.1 | 0.312 |
> | 0.3 | 0.335 |
> | 0.5 | 0.346 |
> | 0.7 | 0.340 |
> | 0.9 | 0.301 |
>
> These results suggest that the method is reasonably robust in a moderate range ($\tau \in \{0.3, 0.5, 0.7\}$), with the best performance at $\tau=0.5$. Performance drops at the extremes, which is consistent with over-merging at low thresholds and over-splitting at high thresholds.
>
> In addition, we also evaluated the method on Qwen3-8B:
>
> ||NQ|TriviaQA|PopQA|HotpotQA|2Wiki|MuSique|Bamboogle|Avg.
> |---|---|---|---|---|---|---|---|---|
> |Search-R1| **0.440**| 0.631| 0.418|0.372|0.355|**0.157**|0.430|0.400|
> |InfoReasoner|0.405|**0.730**|**0.451**|**0.395**|**0.536**|0.152|**0.552**|**0.460**|
>
> InfoReasoner improves the average score from 0.400 to 0.460, confirming that the framework generalizes well to other models.
>
> **W5 & W6: Writing and citation accuracy.**
>
> **Reply**: We have thoroughly proofread the manuscript and carefully fixed citations.

---

> > ### Author Rebuttal · Reviewer_2hVh · 2026-04-03
> >
> > I thank the authors for the rebuttal. I suggest they clarify my first concern (W1) better in the paper.
> > I maintain my positive score and hope to see the paper at ICML.

---

### Official Review · Reviewer_Tocf · 2026-03-11

**Soundness:** 2
**Presentation:** 3
**Significance:** 3
**Originality:** 2
**Overall Recommendation:** 4
**Confidence:** 4

**Summary:**

This paper proposes InfoReasoner, a framework to improve agentic retrieval in Large Reasoning Models by introducing semantic information gain as an intrinsic reward to address sparse and delayed outcome-based feedback. The authors formalise information gain as the reduction of semantic uncertainty over latent answer belief states, proving properties such as non-negativity, telescoping additivity, and channel monotonicity. To estimate belief distributions without annotations, the method samples model outputs and performs bidirectional NLI-based semantic clustering, computing reward as the difference between prior and posterior semantic entropy. Using GRPO training, InfoReasoner achieves up to 5.4% accuracy improvement over baselines like Search-R1 on seven QA benchmarks (Qwen2.5 3B/7B) while reducing retrieval steps, indicating more efficient information-seeking behaviour.

**Compliance With Llm Reviewing Policy:**

Affirmed.

**Final Justification:**

Solved my concern regarding to extended experiments. I have raised my score, thanks!

**Key Questions For Authors:**

1. The paper optimizes agentic retrieval using semantic information gain as an intrinsic reward. However, prior work, such as IGPO (ICLR 2026), already explores information gain–based rewards for training multi-turn LLM agents and search-enhanced reasoning, while EMPO (NeurIPS 2025) proposes fully unsupervised reasoning optimisation via minimising semantic entropy. Could the authors clarify the methodological differences and novelty of InfoReasoner relative to these approaches, and better position the contribution with respect to them?

[1] https://arxiv.org/abs/2510.14967

[2] https://arxiv.org/abs/2504.05812

2. The proposed reward estimation requires sampling M = 12 outputs per context and performing pairwise NLI comparisons (O(M²)) to construct semantic clusters. This may introduce significant computational overhead during RL rollouts. Could the authors clarify the training cost compared to baselines such as Search-R1, e.g., in terms of training time, FLOPs, or rollout cost? Additionally, how scalable is the method when applied to larger models or longer trajectories?

3. Since the intrinsic reward encourages higher agreement among sampled answers, is there a risk of reward hacking or semantic collapse, where the model learns to produce highly consistent but incorrect answers (i.e., confidently wrong outputs) to maximize the reward? Could the authors discuss how the method mitigates this issue and whether such behavior was observed during training?

4. The theoretical analysis (e.g., Proposition 2.6) relies on an idealized Bayesian update assumption, where new evidence dominates the model’s prior belief. However, LLMs are known to exhibit strong prior biases (e.g., sycophancy and confirmation bias). Could the authors clarify how well this assumption holds in practice, and whether such biases may weaken the theoretical guarantees or affect the effectiveness of the proposed reward formulation?

**Limitations:**

yes.

**Strengths And Weaknesses:**

**Soundness**
- Strengths

Overall, the paper is technically solid and well-executed. The experiments use strong and recent baselines (e.g., Search-R1, Search-o1), evaluate on 7 QA datasets, and provide clear algorithms, prompts, and hyperparameters, making the method reasonably transparent and reproducible. The main claims are also well linked to evidence (theory, performance, and efficiency), and the paper includes useful ablations and analyses to support the proposed mechanism.

- Weaknesses

Training cost and fairness are unclear.
The method requires sampling M=12 outputs per context and performing O(M²) NLI comparisons to compute the reward. This likely introduces substantially higher training cost compared to baselines such as Search-R1. However, the paper does not report training FLOPs, GPU hours, or wall-clock time, making it difficult to judge whether the comparisons are fair.

Lack of variance or multi-seed reporting.
The main experimental results (e.g., Table 1) do not report standard deviation or results across multiple random seeds. Given the stochastic nature of RL training, this makes it harder to assess the statistical reliability of the reported gains.

Limited analysis of systematic failure modes.
The paper includes case studies of misleading retrieval and reasoning failures, but it does not provide aggregate statistics on failure patterns. For example, it would be useful to know how often NLI clustering errors lead to incorrect reward signals or degraded training behavior.


**Presentation**

The paper is generally clear and well structured.

**Significance**
- Strengths

The paper tackles an important and timely problem in Agentic RAG—reward sparsity in multi-step retrieval and reasoning. The proposed semantic information gain framework provides a potentially useful way to introduce dense intrinsic rewards, which could influence how future work trains retrieval-based reasoning agents.

- Weaknesses

Limited demonstration of generality.
While the proposed framework appears conceptually transferable, the experiments are limited to QA tasks. It would strengthen the impact if the paper discussed or provided evidence of applicability to other agent settings, such as code generation or tool-augmented reasoning (e.g., math or API use).

**Originality**
- Strengths

The paper introduces a novel perspective by formulating semantic entropy reduction as an intrinsic reward for open-domain agentic retrieval, providing a principled way to densify RL signals. The theoretical analysis—particularly the telescoping additivity property—adds a more formal grounding to what could otherwise be viewed as a heuristic reward design.

- Weaknesses

Positioning relative to closely related work is incomplete.
The paper does not clearly differentiate itself from highly related or concurrent work (e.g., IGPO, which also uses information gain–based rewards). A more explicit comparison would help justify the novelty.

Unclear necessity of the theoretical formulation.
While the telescoping additivity proof strengthens the conceptual framing, it is not fully clear whether this theoretical property is essential for the practical method, or mainly serves as post-hoc justification for the reward design.

---

> ### Author Rebuttal · Authors · 2026-03-29
>
> Dear Reviewer Tocf,
>
> We greatly appreciate your thoughtful feedback.
>
> **W1 & Q2: Training cost and scalability**
>
> **Reply**: We provide the training cost on 8×A800 80GB:
>
> | Quantity | Value |
> |---|---|
> | NO. of steps | 300 |
> | Avg total time / step | 472.2 s |
> | Avg IG time / step | 225.8 s |
> | Avg non-IG time / step | 246.4 s |
> | Wall-clock | 39.35 h |
> | GPU-hours | 314.8 |
>
> The IG computation adds ~1.9× training overhead. However, it is **training-only**, no inference cost, and the learned policy reduces inference cost by 50-67% fewer retrieval turns (Table 4). **For scalability**, the IG cost grows linearly with reasoning steps, while its relative overhead decreases for larger models.
>
> **W2: Multi-seed variance reporting.**
>
> **Reply**: Standard deviations over 3 runs:
>
> ||NQ|TriviaQA|PopQA|HotpotQA|2Wiki|MuSique|Bamboogle|Avg.
> |---|---|---|---|---|---|---|---|---|
> |InfoReasoner-3B| 0.453 (±0.004)| 0.634 (±0.019)| 0.442 (±0.016)|0.344 (±0.008)|0.324 (±0.003)|0.080 (±0.007)|0.144 (±0.005)|0.346 |
> |InfoReasoner-7B|0.447 (±0.003)|0.614 (±0.008)|0.416 (±0.015)|0.414 (±0.017)|0.302 (±0.004)|0.120 (±0.006)|0.424 (±0.014)|0.391|
>
> **W3: Aggregate failure mode analysis and NLI error rates.**
>
> **Reply**: We analyzed 5,000 logged retrieval steps from late training. The joint distribution of IG sign and EM outcome:
>
> | | EM = 1 | EM = 0 | Total |
> |---|---|---|---|
> | IG > 0 | 29.78% | 39.64% | 69.42% |
> | IG < 0 | 13.26% | 16.48% | 29.74% |
> | IG = 0 | 0.32% | 0.52% | 0.84% |
> | Total | 43.36% | 56.64% | 100.00% |
>
> From this distribution, we identify two key patterns: 1) **Dense credit assignment**: 70% of wrong-answer steps (EM=0) still receive positive IG, capturing retrieval utility even when final reasoning fails (Fig. 10). 2) **Reward/outcome mismatch**: 13.26% have IG<0 yet EM=1, arising when retrieval was unnecessary or the reward estimator is imperfect.
>
> The $\lambda$ ablation (Table 2) confirms this: adding IG at $\lambda=0.6$ improves accuracy by 4.7 points over $\lambda=0.0$, while over-emphasizing it ($\lambda=1.0$) degrades from 34.6% to 32.0%, showing dense IG is useful when balanced with final-task supervision.
>
> For NLI clustering errors, we recomputed IG with alternative backbones. The sign agreement with our default DeBERTa-large-MNLI:
>
> | NLI Model | Sign Agreement |
> |---|---|
> | RoBERTa-large-MNLI | 0.883 |
> | DeBERTa-large-MNLI-Fever-ANLI | 0.883 |
> | DistilBART-MNLI-12-3 | 0.850 |
>
> At most ~12–15% of steps receive flipped reward signs due to NLI differences, providing an upper bound on NLI-induced corruption. Combined with EM anchoring and GRPO's group-relative normalization, this noise level is well-tolerated, consistent with stable dynamics in Fig 2a and 2b.
>
> **W4: Generality beyond QA to other agent settings.**
>
> **Reply**: We conduct an additional experiment on **Tool-Integrated Reasoning (TIR)** using MATH500, where the model interleaves reasoning with Python code execution:
>
> | Method | MATH500 |
> |---|---|
> | ZeroTIR-7B (arxiv:2505.07773) | 0.802 |
> | SimpleTIR-7B (arxiv:2509.02479) | 0.846 |
> | InfoReasoner-7B (ours) | **0.858** |
>
> InfoReasoner-7B outperforms both baselines, confirming that the framework transfers to TIR settings beyond QA.
>
> **W5 & Q1: Positioning difference to IGPO and EMPO.**
>
> **Reply**: Compared with IGPO, our reward is defined in **semantic class space** rather than directly on answer probability: we sample outputs, cluster them via bidirectional NLI, and measure how much the retrieved evidence increases the probability of the **gold semantic class** relative to a retrieval-free baseline.
>
> Compared with EMPO, our method is **retrieval-specific** reward anchor by gold answer rather than a general unsupervised reasoning objective, and uses **retrieval-conditioned semantic IG** rather than absolute semantic entropy minimization.
>
> **W6: Role of theoretical properties in the practical method.**
>
> **Reply**:  We agree that the theory is not strictly necessary to implement the method. Its role is to make the reward design principled rather than heuristic: it links step-level IG to global uncertainty reduction, and clarifies why negative realized IG is a useful penalty rather than a failure case.
>
> **Q3: Reward hacking risk.**
>
> **Reply**: Our practical reward is not agreement maximization or pure entropy minimization, it rewards increased probability mass on the **gold semantic class**, so a model that becomes consistently wrong is not rewarded simply for being more self-consistent.
>
> **Q4: Bayesian update assumption and LLM prior biases in practice.**
>
> **Reply**: We agree that this assumption is idealized and mainly introduced for tractability. In practice, prior biases can reduce the fidelity of the reward, so the realized IG may not match the ideal Bayesian quantity exactly. We therefore view the practical reward as an imperfect but still informative proxy for retrieval utility.

---

> > ### Author Rebuttal · Reviewer_Tocf · 2026-04-01
> >
> > Thanks for your informative reply! I solved most of my concerns for both practical usage and detailed experiments. Here, I left my further questions and interesting points:
> > 1. I see the full analysis of failure mode and NLI error rates, and it seems like adding IG at $\lambda=?$ is one of the important factors for this method, so may I ask, do u have any wise way to choose the balanced value from?
> > 2. For generalisation beyond QA, I see good results on math500; however, what I am really curious about is the more complex cases where long-horizon agentic action matters. To be specific, can u try with the recent deep search benchmark, like WebDetective (https://arxiv.org/abs/2510.05137) or Browsecamp (https://arxiv.org/abs/2504.12516) for more interesting analysis? I mean, if we can also get promising results on these, I would be very happy to raise my score.
> > 3. Since you already admitted the flaw of the current theory and assumption, can u propose or change it into a more rational version? I would like to see how this ideality, combined with practical limitations, can produce a more indicative way for our community to continue exploration.

---

> > > ### Author Response · Authors · 2026-04-02
> > >
> > > Dear Reviewer Tocf,
> > >
> > > Thank you for the continued discussion. We address each follow-up question below and welcome any further feedback. Thank you very much!
> > >
> > > **FQ1: How to choose $\lambda$?**
> > >
> > > **Reply**: We selected $\lambda$ via grid search over $\lbrace 0.0, 0.2, 0.4, 0.6, 0.8, 1.0\rbrace$, yielding an inverted-U curve (Table 2) with optimum at $\lambda=0.6$. The surface is smooth, $\lambda \in \lbrace 0.4, 0.6\rbrace$ both substantially outperform the endpoints, so moderate values work without fine-grained tuning.
> > >
> > > Intuitively, $\lambda$ controls the trade-off between *epistemic exploration* (IG rewards uncertainty-reducing retrieval) and *pragmatic exploitation* (EM anchors final-answer correctness). If $\lambda$ is too small, the model receives little dense credit for useful retrieval; if it is too large, the intrinsic term can outweigh the final-task objective.
> > >
> > > In practice, based on our current ablation, $\lambda \in [0.4, 0.6]$ is a reasonable default. An interesting future direction is adaptive $\lambda$ scheduling: higher $\lambda$ for early exploration, decaying toward EM-dominant training as the policy matures.
> > >
> > > **FQ2: Results on long-horizon agentic benchmarks (e.g., WebDetective, BrowseCamp)?**
> > >
> > > **Reply**: Thanks for this constructive suggestion. We evaluate on **WebDetective**, a long-horizon multi-hop deep search benchmark requiring agents to navigate Wikipedia through iterative search and visit actions. The evaluation results are shown below:
> > >
> > > |Model | Knowledge Sufficiency $\uparrow$ | Search Score $\uparrow$ | Generation Score $\uparrow$ | Knowledge Utilization $\uparrow$ | Forget $\downarrow$ | Lead-astray $\downarrow$ | Pass@1 $\uparrow$ |
> > > |---|---|---|---|---|---|---|---|
> > > |GPT-5-Chat|0.580|0.595|**0.157**|**0.281**|0.474|**0.319**|*0.295*|
> > > |Gemini-2.5-Pro|0.655|0.730|0.116|0.247|**0.443**|0.351|0.285|
> > > |DeepSeek-V3.1|0.615|0.565|*0.136*|0.163|*0.447*|0.447|0.170|
> > > |Search-R1-3B-Base|*0.880*|*0.905*|0.110|0.250|0.483|0.364|0.265|
> > > |InfoReasoner-3B (ours)|**0.890**|**0.910**|0.119|*0.269*|0.506|*0.337*|**0.300**|
> > >
> > > Key observations:
> > >
> > > 1. Relative to Search-R1-3B-Base, InfoReasoner **improves Pass@1 from 0.265 to 0.300 (+3.5pp)**. Notably, its Pass@1 is also **competitive with much larger frontier models** in this benchmark, such as GPT-5-Chat (0.295) and Gemini-2.5-Pro (0.285), suggesting that improved step-level retrieval credit assignment can materially benefit long-horizon search even at a small model scale.
> > > 2. Compared with Search-R1-3B-Base, InfoReasoner also achieves higher Knowledge Sufficiency (0.890 vs. 0.880), Search Score (0.910 vs. 0.905), Knowledge Utilization (0.269 vs. 0.250), and lower Lead-astray (0.337 vs. 0.364), suggesting that IG rewards improve evidence seeking and reduce misleading retrieval in long-horizon settings.
> > >
> > > Overall, these results confirm that the semantic IG reward transfers effectively to long-horizon agentic deep search, yielding strong retrieval quality and competitive accuracy even against frontier models.
> > >
> > > *Note: Knowledge Sufficiency measures whether the agent retrieved enough evidence; Search Score measures retrieval quality; Generation Score combines knowledge utilization and refusal behavior; Knowledge Utilization measures correct use of sufficient evidence; Forget captures cases where sufficient evidence was retrieved but unused; Lead-astray captures cases where correct evidence was available but the agent answered incorrectly. The results of GPT-5-Chat, Gemini-2.5-Pro, and DeepSeek-V3.1 are from the original paper.*
> > >
> > >
> > >
> > >
> > > **FQ3: Can you propose a more realistic theoretical formulation?**
> > >
> > > **Reply**: This is a very constructive suggestion. A more realistic extension is to replace the idealized update with a **tempered Bayesian update**:
> > >
> > > $$b_{t+1}^{(\alpha)}(y) \propto P(O_t \mid Y=y, a_t)^{\alpha} \, b_t(y),$$
> > >
> > > where $\alpha \in (0,1]$ controls how strongly the model incorporates new evidence. When $\alpha=1$, this recovers the idealized Bayesian update used in our analysis; when $\alpha<1$, it captures partial grounding, i.e., the model under-weights retrieved evidence relative to its prior.
> > >
> > > Under this view, the practical IG reward should be interpreted not as an exact realization of ideal Bayesian information gain, but as a tractable proxy for uncertainty reduction under imperfect evidence integration. This also offers one possible interpretation of our empirical finding that a moderate $\lambda$ works best: when the intrinsic signal is imperfect, it should complement rather than dominate the final-task reward.
> > >
> > > We believe this **partial-grounding** perspective is a more realistic direction for future theory, because it connects the idealized analysis to practical limitations while suggesting a concrete path forward for the community. An interesting future direction is to estimate or adapt $\alpha$ from the model’s actual evidence-use behavior.

---

### Official Review · Reviewer_vjbg · 2026-03-13

**Soundness:** 4
**Presentation:** 3
**Significance:** 3
**Originality:** 3
**Overall Recommendation:** 4
**Confidence:** 4

**Summary:**

The research addresses a critical problem in retrieval-augmented large reasoning models: the lack of dense, principled reward signals for optimizing retrieval actions during agentic reasoning. The authors propose InfoReasoner, a framework that introduces a synthetic semantic information gain reward to incentivize effective information seeking. Theoretically, they redefine IG as uncertainty reduction over belief states, proving properties like non-negativity and telescoping additivity. Practically, they design an output-aware intrinsic estimator using bidirectional textual entailment to compute IG without manual annotations. Experiments on seven QA benchmarks show consistent improvements over baselines (up to 5.4% accuracy gain).

**Compliance With Llm Reviewing Policy:**

Affirmed.

**Final Justification:**

My concerns have been addressed and I maintain my positive score.

**Key Questions For Authors:**

1. How does the choice of NLI model affect IG estimation precision versus simpler methods like embeddings, and how does NLI noise specifically impact the convergence stability of GRPO?
2. How does the framework handle conflicting information within the retrieval corpus, and can it distinguish between genuine uncertainty reduction and false confidence caused by outdated or noisy data?
3. Can the IG reward mechanism be generalized to multimodal settings?

**Limitations:**

Yes

**Strengths And Weaknesses:**

**Strengths:**

1. The paper establishes a rigorous theoretical foundation by modeling agentic retrieval as a POMDP, which enables the generation of dense, intrinsic IG rewards to effectively address the sparse reward challenge.
2. The semantic clustering approach leveraging bidirectional entailment is a good design that allows for accurate uncertainty estimation and bypasses the need for manual retrieval annotations.
3. InfoReasoner demonstrates high resource efficiency by learning to invoke retrieval only when necessary, consistently outperforming RL and SFT baselines in both accuracy and retrieval turn economy across multiple benchmarks.

**Weaknesses:**

1. The framework's heavy dependence on pretrained NLI models for semantic clustering poses a risk where model **biases or failures** to capture subtle linguistic nuances can propagate errors into the IG reward signals.
2. The requirement for sampling multiple trajectories and performing repetitive NLI checks significantly increases computational overhead, which may limit the scalability of the approach for long chain reasoning or larger LRM training.
3. The work lacks a discussion on how the method handles hallucinated retrieval results that might falsely appear to reduce uncertainty. Furthermore, the evaluation is primarily focused on QA, leaving its generalizability to complex agentic reasoning tasks unverified.

---

> ### Author Rebuttal · Authors · 2026-03-29
>
> Dear Reviewer vjbg,
>
> We greatly appreciate your thoughtful feedback. Below, we respond to your concerns and questions in detail.
>
> **W1: NLI model robustness and potential bias propagation into IG rewards.**
>
> **Reply**: Our design uses **bidirectional** entailment for semantic clustering, which is stricter than one-way matching or embedding similarity and reduces false-positive grouping. We use DeBERTa-large, a strong NLI model [arXiv:2006.03654]:
>
> | Model | MNLI-m Acc | SQuAD F1 |
> |---|---|---|
> | XLNet-large | 90.8 | 95.1 |
> | Megatron-336M | 89.7 | 94.2 |
> | **DeBERTa-large** | **91.1** | **95.5** |
>
> **At the reward level**, since the same NLI model is used for both prior and posterior estimation, shared biases tend to cancel out in the relative IG computation. Empirically, training dynamics (Fig. 2) show stable EM improvement with no signs of reward corruption from NLI noise.
>
> **W2: Computational overhead and scalability of the reward estimation.**
>
> **Reply**: We agree that the semantic information gain (IG) reward introduces additional training-time. On 8×A800 80GB, we measure the following cost:
>
> | Quantity | Value |
> |:--|---|
> | Number of steps | 300  |
> | Avg total time / step | 472.2 s |
> | Avg IG time / step | 225.8 s |
> | Avg non-IG time / step  | 246.4 s |
> | Wall-clock | 39.35 h |
> | GPU-hours | 314.8 |
>
> The non-IG time per step (246.4 s) would remain under an outcome-only baseline (e.g., Search-R1), while the IG computation adds 225.8 s on top. This is a **training-only** overhead, no additional costs at inference time, and the learned policy can even reduce inference cost by using 50-67% fewer retrieval turns (Table 4).
>
> In terms of scalability, for fixed reward-estimation settings, the additional IG cost grows approximately linearly with the number of reasoning steps, and its *relative* overhead decreases for larger policy models because the cost of the base model grows with model size while the auxiliary IG computation remains comparatively stable.
>
> **W3: Hallucinated retrieval handling and generalizability beyond QA.**
>
> **Reply**: Regarding hallucinated or misleading retrieval, our framework has a built-in mechanism for detecting and penalizing such cases. When retrieval returns irrelevant or misleading content, the model's posterior belief becomes *more dispersed* rather than concentrated, resulting in a *negative* IG. As stated in our Remark after Proposition 2.6, "a negative reward discourages the policy from executing retrieval actions that confuse the model."
>
> Furthermore, even if noisy data causes the model to converge on a *wrong* answer with high confidence, the IG reward based on the *gold semantic class* $c^\star$ (Eq. 17) will remain low or negative, since $\log p(c^\star \mid C_t)$ does not increase when the model is confidently wrong. Thus, the framework does not reward mere consensus, **it rewards consensus on the *correct* answer**.
>
> Regarding generalizability, we conduct an additional experiment on **Tool-Integrated Reasoning (TIR)** using MATH500, where the agent interleaves reasoning with Python code execution. We adapt the IG reward by estimating the prior belief before code execution and the posterior after the interpreter returns results:
>
> | Method | MATH500 |
> |---|---|
> | ZeroTIR-7B (NeurIPS'25) | 0.802 |
> | SimpleTIR-7B (ICLR'26) | 0.846 |
> | InfoReasoner-7B (ours) | **0.858** |
>
> InfoReasoner-7B outperforms both baselines, suggesting that the semantic IG reward can generalize to more complex agentic tasks.
>
> **Q1: NLI vs. embeddings for IG precision; NLI noise impact on GRPO stability.**
>
> **Reply**: NLI captures semantic *equivalence* while embeddings only capture *similarity*, so embeddings can over-merge related but non-equivalent answers, reducing clustering precision.
>
> Regarding GRPO stability: NLI noise is mitigated because 1) IG is a relative prior-vs.-posterior difference, so shared NLI bias will partially cancel, 2) GRPO uses group-relative normalization, and (3) the EM reward provides an additional anchor. Empirically, the training curves shows EM evolves smoothly without obvious oscillation or collapse.
>
> **Q2: Conflicting retrieval information and genuine vs. false uncertainty reduction.**
>
> **Reply**: Our IG estimator (Eq. 17) tracks probability mass on the *correct* semantic class $c^\star$, not general confidence. When conflicting evidence disperses the model’s answers or shifts probability away from $c^\star$, the reward naturally decreases. Similarly, if noisy evidence induces false confidence in a wrong answer, $p(c^\star \mid C_t)$ does not increase, so the reward remains low. Any residual mismatch is further checked by the EM reward.
>
> **Q3: Generalizability to multimodal settings.**
>
> **Reply**: Our method does not require a specific input modality. As long as the model outputs texts, the reward mechanism remains compatible.

---

> > ### Author Rebuttal · Reviewer_vjbg · 2026-04-02
> >
> > Thank you for the rebuttal. My concerns have been addressed and I maintain my positive score.

---

### Decision · Program_Chairs · 2026-04-30

**Decision:**

Accept (regular)

**Comment:**

This paper proposes **InfoReasoner**, a framework for improving agentic retrieval in large reasoning models by introducing semantic information gain (IG) as an intrinsic reward. The method formulates retrieval as uncertainty reduction over latent belief states and estimates IG via sampling and NLI-based semantic clustering, enabling dense reward signals without manual annotation. Experiments across multiple QA benchmarks show consistent gains in both accuracy and retrieval efficiency.

Reviewers generally find the paper technically sound and well-motivated. Strengths include a principled formulation of retrieval as uncertainty reduction, a practical reward design without step-level supervision, and solid empirical improvements across datasets and model scales. Weaknesses noted by reviewers include reliance on NLI models (with potential bias), increased training cost due to sampling and pairwise comparisons, limited evaluation beyond QA tasks, and some gaps in reporting (e.g., training cost, variance, and positioning relative to prior work).

In the rebuttal, the authors provide additional experiments and clarifications addressing most concerns, including detailed training cost analysis, robustness to NLI choices, extended evaluations on tool-integrated and long-horizon tasks, and clearer positioning of their contributions. Reviewers acknowledge that their concerns have been largely or fully addressed and maintain positive assessments.

Overall, while some limitations remain, the paper makes a meaningful contribution to reward design for agentic retrieval. I recommend acceptance.